# Somatic role of SYCE2: an insulator that dissociates HP1α from H3K9me3 and potentiates DNA repair

Noriko Hosoya ⓘ, Masato Ono, Kiyoshi Miyagawa

**The synaptonemal complex is a proteinaceous structure essential for meiotic recombination, and its components have been assumed to play a role exclusively in the germ line. However, SYCE2, a component constituting the synaptonemal complex, is expressed at varying levels in somatic cells. Considering its potent protein-binding activities, it may be possible that SYCE2 plays a somatic role by affecting nuclear functions. Here, we show that SYCE2 constitutively insulates HP1α from trimethylated histone H3 lysine 9 (H3K9me3) to promote DNA double-strand break repair. Unlike other HP1α-binding proteins, which use the canonical PXVXL motifs for their bindings, SYCE2 interacts with the chromoshadow domain of HP1α through its N-terminal hydrophobic sequence. SYCE2 reduces HP1α-H3K9me3 binding without affecting H3K9me3 levels and potentiates ataxia telangiectasia mutated–mediated double-strand break repair activity even in the absence of exogenous DNA damage. Such a somatic role of SYCE2 is ubiquitously observed even if its expression levels are low. These findings suggest that SYCE2 plays a somatic role in the link between the nuclear microenvironment and the DNA damage response potentials as a scaffold of HP1α localization.**

## Introduction

Meiosis is a cell division process unique to germ cells and possesses some specific features distinct from mitosis. The synaptonemal complex is a meiosis-specific supramolecular proteinaceous structure that is formed between the paternal and maternal chromosomes (Page & Hawley, 2004). The synaptonemal complex consists of two parallel axial/lateral elements, which colocalize with the sister chromatids of each homolog along with a central element, and transversal filaments, which connect the two axial/lateral elements and the central element along their entire length during meiotic prophase I. The axial/lateral elements are encoded by the meiosis-specific synaptonemal complex proteins SYCP2 and SYCP3. Transversal filaments are encoded by SYCP1, and the central elements are encoded by SYCE1, SYCE2, SYCE3, and TEX12 (Page &

Hawley, 2004; Hamer et al, 2006; Schramm et al, 2011). Although the components of the synaptonemal complex were first considered to be expressed only in the germ line, some of them are reported to be expressed in various somatic tumors by a demethylation-dependent process (Türeci et al, 1998; Lim et al, 1999; Niemeyer et al, 2003; Simpson et al, 2005; Kang et al, 2010). The roles of synaptonemal complex proteins in somatic cells are not well understood, except for the role of SYCP3 reported by our group (Hosoya et al, 2012). We reported that SYCP3 interferes with the BRCA2 tumor suppressor and inhibits the intrinsic homologous recombination (HR) pathway, indicating the role of a synaptonemal complex protein in regulating the DNA damage response and repair of DNA double-strand breaks (DSBs).

The DNA damage response and repair of DSBs play a central role in the maintenance of genome integrity. The early steps of the signaling cascade involve sensing of the DSBs by the ataxia telangiectasia mutated (ATM) kinase, followed by subsequent recruitment of the DNA repair factors and initiation of the repair process. DSBs are predominantly repaired by either non-homologous end joining (NHEJ) or HR. NHEJ is an error-prone repair pathway that is mediated by the direct joining of the two broken ends, whereas HR is an error-free repair pathway that requires a non-damaged sister chromatid to serve as a template for repair.

Increasing evidence suggests that the nuclear architecture, including chromatin states, is important for the regulation of the DNA damage response and repair. Among the number of different chromatin states that have currently been annotated (Ernst & Kellis, 2010; Filion et al, 2010), heterochromatin and euchromatin are the two classical broad divisions of chromatin states (Maison & Almouzni, 2004). Heterochromatin was originally described as a region in the nucleus which is densely stained with DAPI and corresponds to a highly compacted form of chromatin. Conversely, the euchromatin region is weakly stained with DAPI and less compacted. A specific histone mark, the trimethylation of histone H3 on lysine 9 (H3K9me3), is known to be enriched in heterochromatin. This histone mark can be bound by specific non-histone proteins that can change the nuclear environments. Among these proteins, heterochromatin protein 1 (HP1) is the key factor for the establishment and maintenance of heterochromatin. This protein has two conserved domains: the N-terminal chromodomain and

Laboratory of Molecular Radiology, Center for Disease Biology and Integrative Medicine, Graduate School of Medicine, The University of Tokyo, Tokyo, Japan

Correspondence: nhosoya-tky@umin.net

the C-terminal chromoshadow domain connected by an intervening region or hinge region. The chromodomain of HP1 directly interacts with H3K9me3, which is crucial for the maintenance of the heterochromatic state (Bannister et al, 2001; Lachner et al, 2001). The intervening region, or alternatively, the hinge region, interacts with RNA and DNA (Muchardt et al, 2002; Meehan et al, 2003), and the chromoshadow domain is involved in HP1 dimerization and protein–protein interactions (Nielsen et al, 2001; Thiru et al, 2004). In mammalian cells, there are three HP1 variants: HP1α, HP1β, and HP1γ. They exhibit distinct subnuclear localization patterns: HP1α and HP1β primarily associate with heterochromatic regions of the genome, whereas HP1γ largely localizes to euchromatic regions (Maison & Almouzni, 2004).

Originally identified as a critical component of heterochromatin (Eissenberg et al, 1990), HP1 has recently been recognized to function in the DNA damage response and repair. Knockdown of all three variants of HP1 has been shown to alleviate the requirement of ATM kinase for heterochromatic DSB repair (Goodarzi et al, 2008). HP1β is rapidly phosphorylated at threonine 51 by the DNA-damage responsive enzyme casein kinase 2, resulting in dissociation of HP1β from H3K9me3, which promotes the H2AX phosphorylation that triggers DNA repair (Ayoub et al, 2008). Subsequently, HP1β was shown to exhibit two distinct behaviours at DSB sites: rapid and transient mobilization that was most evident in the heterochromatic regions, followed by slower recruitment (Ayoub et al, 2009; Luijsterburg et al, 2009; Zarebski et al, 2009). Moreover, the DNA damage-induced displacement of HP1β from H3K9me3 was reported to be required for the binding of the Tip60 histone acetyltransferase to H3K9me3 (Sun et al, 2009). Phosphorylation of the co-repressor Kruppel-associated box domain–associated protein 1 serine 473 by the Chk2 cell cycle checkpoint kinase was also shown to promote the mobilization of HP1β from heterochromatin and subsequent DNA repair (Bolderson et al, 2012). In addition, HP1α has also been reported to promote HR (Baldeyron et al, 2011; Alagoz et al, 2015).

Dissociation of HP1 from H3K9me3 has also been shown to play roles in other cellular functions. Histone H3 serine 10 phosphorylation by Aurora B causes HP1 dissociation from mitotic heterochromatin, indicating the essential role of HP1 in accurate chromosome segregation in somatic cells (Hirota et al, 2005). The HP1α-binding protein POGZ was shown to cause the dissociation of HP1α from mitotic chromosome arms and dissociation of Aurora B kinase from chromosome arms during M phase (Nozawa et al, 2010). The DYRK1A kinase was reported to phosphorylate histone H3 threonine 45 and serine 57 to regulate the binding of HP1 isoforms and antagonize HP1-mediated transcriptional repression (Jang et al, 2014). The SUMO protease SENP7 was recently shown to interact directly with HP1α and to enable HP1α retention and accumulation at pericentric heterochromatin without affecting H3K9me3 levels (Maison et al, 2012; Romeo et al, 2015). The nuclear oncogene SET was shown to control DNA repair by retention of HP1 to chromatin (Kalousi et al, 2015).

In this study, we investigated the somatic roles of SYCE2 that have not been elucidated yet, whereas a previous article reported that the knockout of SYCE2 in mice resulted in less efficient DSB repair in the germ line (Bolcun-Filas et al, 2007). We show here that SYCE2 is expressed at varying elevated levels in somatic cancer cells in a demethylation-dependent manner and at very low levels in normal cells. We also found that SYCE2 potentiates ATM activity and ATM-mediated DNA DSB repair, and eventually induces resistance to DNA damage. Moreover, we found that SYCE2 directly binds to HP1α and reduces the binding of HP1α to H3K9me3. By showing that the mutant cells expressing SYCE2 lacking the binding site to HP1α cannot potentiate ATM activity, we conclude that the dissociation of HP1α from H3K9me3 by the insulator SYCE2 is crucial for activating ATM-mediated DNA repair. Taken together, these findings suggest that SYCE2 might be a prime candidate target for treatment of SYCE2-expressing tumors.

# Results

## SYCE2 is expressed at varying elevated levels in somatic cancer cells

We first examined *SYCE2* expression in somatic cells by quantitative real-time RT–PCR analysis using human cancer cell lines from various tissues in addition to normal cell lines and normal human testis (Fig S1A and B). High expression of *SYCE2* was observed in the normal testis, which is in agreement with previous findings (Costa et al, 2005). Although *SYCE2* expression was not expected in non-meiotic cells, aberrant expression of *SYCE2* was detected in various cancer cell lines and at varying low levels in normal cell lines. Particularly, high levels of expression were observed in the T-lymphocyte cell lines Jurkat and Molt4; the erythroblastic cell line K562; the monocytic cell line U937; the breast cancer cell lines MCF7, MDA-MB468, and T47D; and the brain tumor cell line A172. In contrast, normal retinal pigmented epithelial (RPE) cells or normal human mammary epithelial cells showed extremely low levels of expression, whereas human embryonic kidney (HEK293) cells, which are not coming from tumors but are transformed with adenovirus type 5 E1A and E1B, showed low levels but slightly higher than the levels in RPE and human mammary epithelial cells.

We further examined the expression of *SYCE2* in primary tumors by RT–PCR analysis using commercially available samples and found *SYCE2* expression in the cervix, ovary, thyroid, and uterus tumor samples, and high expression in the normal testis. We also performed RT–PCR analysis using three sets of paired samples of patient-matched tumors and normal adjacent tissues. Although *SYCE2* expression was detected in kidney and stomach cancers, as well as in lymphoma samples, no expression was detected in normal tissues from the same patients (Fig S1C). Taken together, these results demonstrate that *SYCE2* is expressed at very low levels in normal cells and at varying elevated levels in tumors from various tissues.

Moreover, induction of *SYCE2* expression was observed in the DLD1 and HT1080 cancer cell lines after treatment with the demethylating agent 5-azacytidine (Fig S1D), indicating that *SYCE2* expression is regulated by a demethylation-dependent process, as has been described for some cancer-testis antigens (Simpson et al, 2005).

## SYCE2 expression leads to resistance to DNA-damaging agents

To investigate the role of SYCE2 expression in somatic cells, we examined whether the changes in SYCE2 expression levels would

affect the sensitivity of the cells to DNA damage. We measured the ability of the cells to form colonies following exposure to ionizing radiation or cisplatin which produces both interstrand and intra-strand cross-links. Knocking down the endogenous *SYCE2* mRNA in MCF7 cells increased the sensitivities to x-ray irradiation and cisplatin (Figs 1A and B, and S1E). Conversely, we also expressed the *SYCE2* gene tagged with the *FLAG* cDNA at its N-terminus in telomerase-immortalized RPE cells and established two inde-pendent clones stably expressing exogenous FLAG-SYCE2 at the levels comparable with those in hematopoietic tumor cells (Fig S1B and F). Unlike mock cells, the FLAG-SYCE2-expressing RPE cells exhibited resistance to x-ray irradiation and cisplatin (Fig 1C and D). These findings indicate that the expression of SYCE2 induces re-sistance to DNA damage.

## DNA damage is less accumulated in SYCE2-expressing cells

We next analyzed the nuclear foci formation of the phosphorylated form of histone H2AX (γH2AX), which is recruited to DSBs in re-sponse to DNA damage (Rogakou et al, 1999). Immunofluorescence detection of the nuclear foci of γH2AX revealed that silencing of SYCE2 in MCF7 cells significantly increases the frequencies of γH2AX foci-positive cells (Fig 2A and B). In the absence of exogenous DNA damage (0 Gy), the frequency increased from 17.7 ± 2.0% (mean ± SD) in control cells to 24.7 ± 4.0% in the MCF7 cells transfected with

*SYCE2*-targeting siRNA. At 1 h after 1-Gy irradiation, the frequency of foci-positive cells in both cell types was 100.0 ± 0.0%. At 8 and 24 h after 1-Gy irradiation, the frequencies of foci-positive cells in control cells were 27.0 ± 2.6% and 17.3 ± 2.5%, respectively, whereas those in the MCF7 cells transfected with *SYCE2*-targeting siRNA were elevated to 46.7 ± 3.1% and 35.0 ± 4.6%, respectively. These results indicate that DNA DSBs are less accumulated in SYCE2-expressing cells.

## SYCE2 expression activates the DSB repair pathways

Because DNA DSBs are less accumulated in SYCE2-expressing cells, we hypothesized that the DSBs are more efficiently repaired in SYCE2-expressing cells. We therefore assessed whether SYCE2 ex-pression affects the major repair pathways for DSBs, NHEJ, and HR.

Activation of the 53BP1 protein contributes to the choice of the DSB repair pathways by promoting NHEJ (Zimmermann & de Lange, 2014). The silencing of SYCE2 in MCF7 cells reduced the radiation-induced foci formation of 53BP1 from 76.6 ± 6.2% to 56.2 ± 6.6% (Fig 2C and D). Moreover, the silencing of SYCE2 in MCF7 cells also showed a tendency to reduce the foci formation of 53BP1 in the steady state from 21.1 ± 8.3% to 15.0 ± 1.9%. Conversely, after stable FLAG-SYCE2 expression, the frequency of radiation-induced 53BP1 foci increased from 68.7 ± 1.9% in mock cells to 90.7 ± 1.5% and 90.9 ± 1.0% in the two FLAG-SYCE2–expressing RPE clones (Fig 2E).

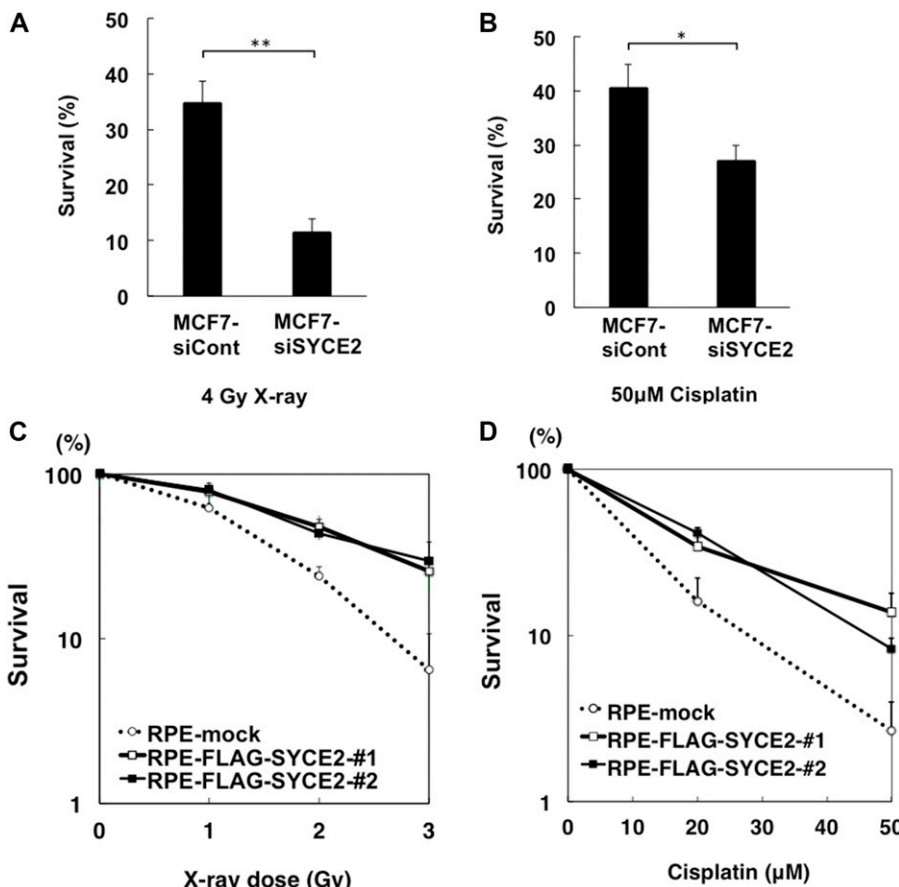

**Figure 1. SYCE2 induces resistance to ionizing radiation and cisplatin in somatic cells.**
**(A, B)** Colony survival of MCF7 cells transfected with nontargeting control siRNA and *SYCE2*-targeting siRNA after 4-Gy x-ray irradiation (A) and treatment with 50 μM cisplatin (B). **(C, D)** Sensitivities of the mock cells and FLAG-SYCE2–expressing RPE cells after treatment with ionizing radiation (C) and cisplatin (D).

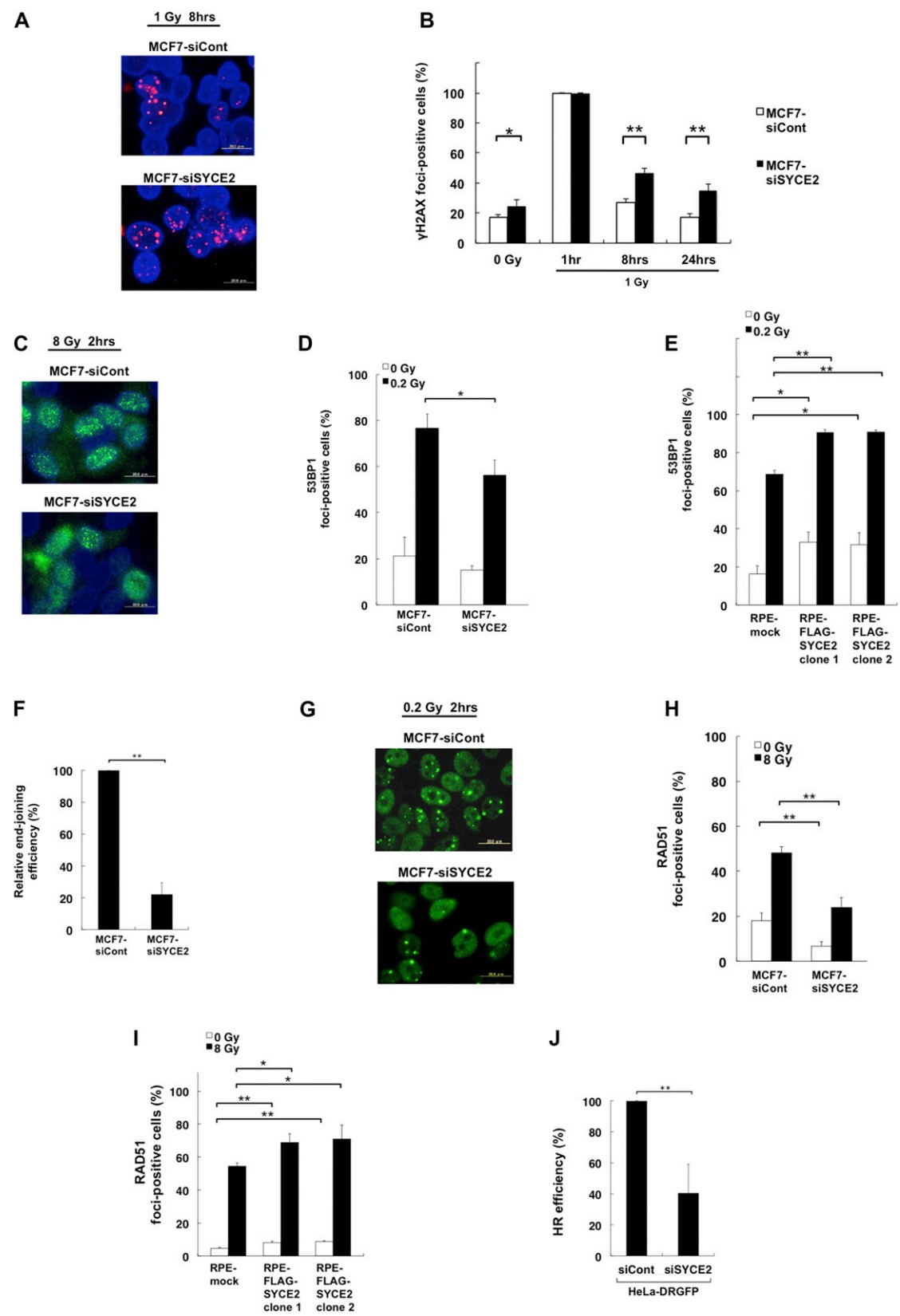

Moreover, stable expression of FLAG-SYCE2 also increased the frequency of 53BP1 foci in the steady state from 16.2 ± 4.4% in mock cells to 32.9 ± 5.2% and 31.5 ± 1.0% in the two FLAG-SYCE2–expressing RPE clones. Supporting these results, the silencing of SYCE2 in MCF7 cells reduced the relative end-joining frequency to 22.2 ± 7.1% compared with that of control cells in the DNA-ligation assay previously described by Buck et al (2006) (Figs 2F and S2A), indicating that NHEJ is promoted by SYCE2 expression. The percentages of the cells containing inaccurate junctions with long nucleotide deletions were 16.7% in both types of cells, suggesting that the fidelity of NHEJ was not affected by SYCE2.

HR regulates the sensitivity to both radiation and DNA cross-linking agents (Nojima et al, 2005). Because somatic SYCE2 expression induced resistance to both ionizing radiation and cisplatin as described above (Figs 1A–D), we hypothesized that HR may also be activated by SYCE2 expression. In HR, RAD51 plays a central role by forming nuclear foci in a DNA damage-dependent manner (Tashiro et al, 2000). We therefore analyzed the radiation-induced foci formation of RAD51 (Fig 2G and H). The silencing of SYCE2 in MCF7 cells reduced the radiation-induced foci formation of RAD51 from 48.3 ± 2.5% to 24.0 ± 4.4%. Moreover, the silencing of SYCE2 in MCF7 cells also reduced the foci formation of RAD51 in the steady state from 18.0 ± 3.5% to 6.7 ± 2.1%. Conversely, stable FLAG-SYCE2 expression increased the radiation-induced frequency of RAD51 foci from 54.3 ± 2.1% in mock cells to 69.0 ± 5.3% and 71.0 ± 8.5% in the two FLAG-SYCE2–expressing RPE clones (Fig 2I). Moreover, the foci formation of RAD51 in the steady state also increased from 4.7 ± 0.6% in mock cells to 8.0 ± 1.0% and 8.7 ± 0.6% in the two FLAG-SYCE2–expressing RPE clones. The total protein levels of RAD51 were not affected by FLAG-SYCE2 expression (Fig S2B). Because the cell cycle distribution was similar between the mock and FLAG-SYCE2–expressing cells both in the steady state and at 2 h after 8-Gy irradiation (Fig S2C), the increased RAD51 foci formation in SYCE2-expressing cells might not have been due to an increase in the number of cells in the S and G2 phases. We also analyzed the foci formation of BRCA1, which is indirectly associated with RAD51 (Fig S2D). The silencing of SYCE2 in MCF7 cells significantly reduced the radiation-induced foci formation of BRCA1 from 61.0 ± 6.2% to 47.3 ± 3.1%. Moreover, the silencing of SYCE2 in MCF7 cells also reduced the foci formation of BRCA1 in the steady state from 34.0 ± 8.5% to 18.7 ± 3.1%. Conversely, stable FLAG-SYCE2 expression increased the frequency of BRCA1 foci from 42.7 ± 3.1% in mock cells to 57.3 ± 4.2% and 61.0 ± 3.6% in the two FLAG-SYCE2–expressing RPE clones

(Fig S2E). Moreover, the foci formation of BRCA1 in the steady state also increased from 15.3 ± 3.1% in mock cells to 34.0 ± 4.0% and 37.0 ± 5.0% in the two FLAG-SYCE2–expressing RPE clones. These results support the idea that SYCE2 expression activates the HR pathway. We therefore measured the effect of SYCE2 expression on HR efficiency with a direct repeat GFP (DR-GFP) assay (Pierce & Jasin, 2005). We assessed the effect of knockdown of endogenous SYCE2 in HeLa-DRGFP cells (Sakamoto et al, 2007). Silencing endogenous SYCE2 in these cells significantly reduced the HR efficiencies to 40.5 ± 18.6% compared with that of control cells (Fig 2J). These findings provide direct evidence that SYCE2 promotes the intrinsic HR activity of somatic cells.

### SYCE2 potentiates ATM activity even in the absence of exogenous DNA damage

The above finding that somatic expression of SYCE2 leads to activated DNA DSB repair led us to hypothesize that SYCE2 expression might activate the sensor protein ATM involved in the early steps of the DNA damage response. In response to DSBs, the MRE11-RAD50-NBS1 complex senses and binds to DSB sites, and recruits and activates the ATM kinase through its autophosphorylation (Bakkenist & Kastan, 2003; Lee & Paull, 2005). Once activated, ATM phosphorylates a large number of downstream proteins (Matsuoka et al, 2007). We therefore investigated the levels of ATM phosphorylation on Ser 1981, a molecular marker of ATM activity, and the protein levels. Silencing of SYCE2 in MCF7 cells decreased autophosphorylation of ATM on Ser 1981, whereas no differences were observed in the levels of ATM expression (Fig 3A). The FLAG-SYCE2–expressing cells showed increased radiation-induced autophosphorylation of ATM on Ser 1981 compared with mock cells (Fig 3B). These results indicate that SYCE2 expression induces ATM activation in response to DNA damage.

To confirm this, we also performed immunofluorescence to assess the effect of SYCE2 expression on ATM autophosphorylation. The silencing of SYCE2 in MCF7 cells using a second siRNA targeting the sequence in 3′ UTR of SYCE2 reduced the radiation-induced foci formation of phosphorylated ATM (Ser 1981) from 13.7 ± 1.7% to 8.9 ± 0.6%. This was rescued to 18.0 ± 3.7% by expressing the siRNA-resistant FLAG-SYCE2 construct, which strengthened the specificity of the observed phenotypes (Figs 3C and D, and S3A and B). Furthermore, the silencing of SYCE2 also reduced the frequencies of the foci formation in the steady state from 6.6 ± 0.5% to 1.8 ± 0.3%, which was rescued to 5.6 ± 1.4% by expressing the siRNA-resistant

**Figure 2. Expression of SYCE2 activates the DSB repair pathway.**
**(A)** Immunofluorescence visualization of γH2AX foci (red) in MCF7 cells transfected with nontargeting control siRNA (upper panel) and those transfected with SYCE2-targeting siRNA (lower panel). Scale bar, 20 μm. **(B)** Percentages of cells containing more than five large γH2AX foci in MCF7 cells transfected with nontargeting control siRNA and those transfected with SYCE2-targeting siRNA, in unirradiated states and at 1, 8, and 24 h after 1-Gy x-ray irradition. A total of 100 cells were examined for each cell clone and at each time point. Columns and bars represent the mean of three independent experiments and SD, respectively. **(C)** Immunofluorescence visualization of 53BP1 foci (green) in MCF7 cells transfected with nontargeting control siRNA (upper panel) and those transfected with siRNA for SYCE2 (lower panel), treated at 2 h after 0.2-Gy x-ray irradiation. Scale bar, 20 μm. **(D, E)** Percentages of cells containing three or more large 53BP1 foci in unirradiated states and at 2 h after 0.2-Gy x-ray irradiation. Columns and bars represent the mean of three independent experiments and SD, respectively. A total of 100 cells were examined for each cell line. **(F)** NHEJ activity measured by in vivo DNA-ligation assay. The relative percentages of recirculation in MCF7 cells transfected with SYCE2-targeting siRNA compared with those in control cells were evaluated in three independent experiments. Columns and bars represent the mean and SD. **(G)** Immunofluorescence visualization of RAD51 foci (green) in MCF7 cells transfected with nontargeting control siRNA (upper panel) and in MCF7 cells transfected with siRNA for SYCE2 (lower panel), stained at 2 h after 8-Gy x-ray irradiation. Scale bar, 20 μm. **(H, I)** Percentages of cells containing more than five large RAD51 foci. Columns and bars represent the mean of three independent experiments and SD, respectively. A total of 100 cells were examined for each cell line. **(J)** HR repair activity measured by the DR-GFP assay. The relative percentages of the GFP-positive cells in HeLa-DRGFP cells transfected with SYCE2-targeting siRNA compared to those transfected with nontargeting control siRNA were evaluated in three independent experiments. Columns and bars represent the mean and SD, respectively.

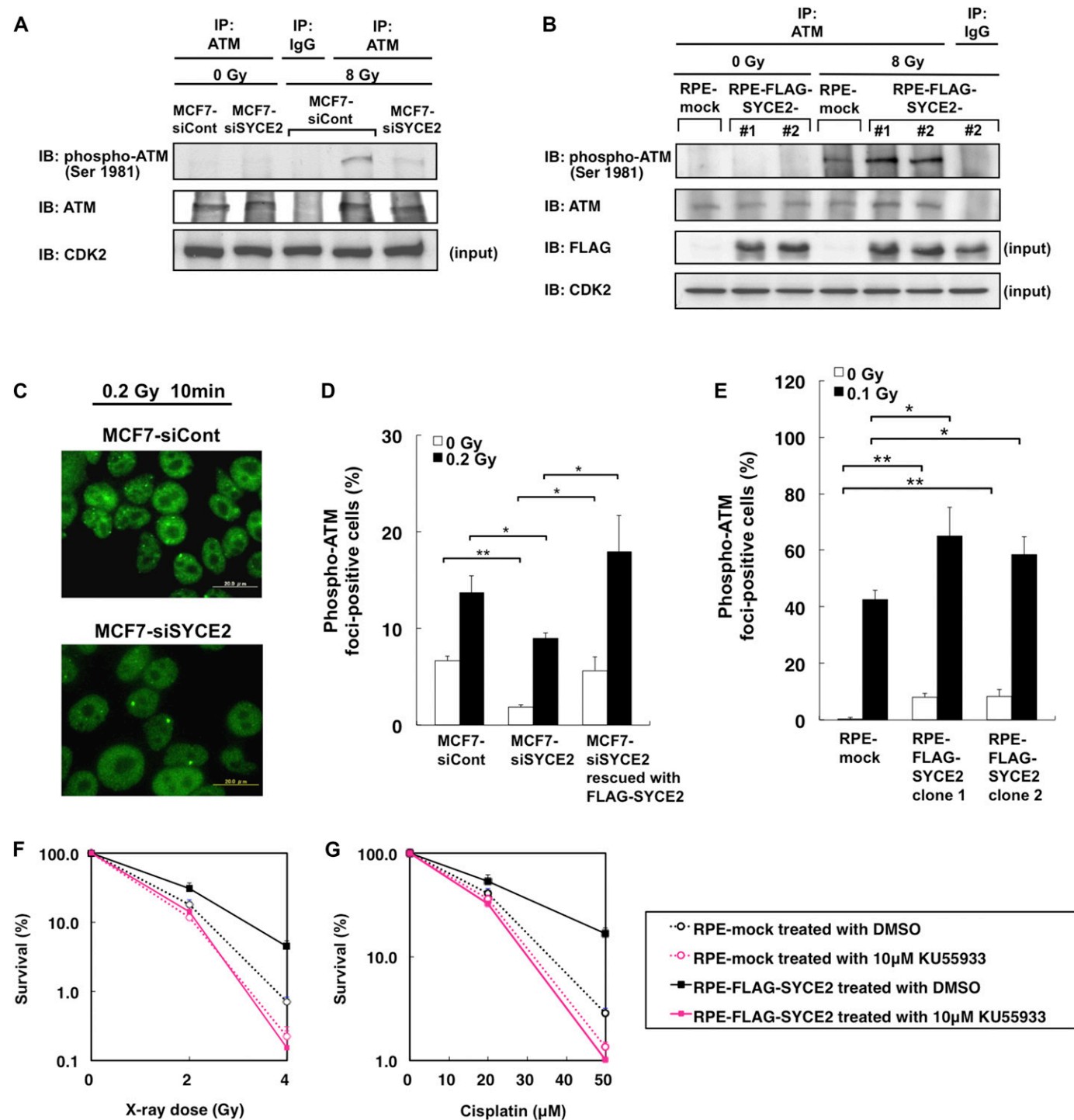

**Figure 3. SYCE2 potentiates the steady-state ATM activity.**
**(A, B)** Immunoprecipitation/Western blot analysis showing the levels of autophosphorylation of ATM on Ser 1981 and ATM expression. 500 μg of total cell lysates of mock cells and RPE cells expressing FLAG-SYCE2 (A) or MCF7 cells transfected with a nontargeting siRNA control or *SYCE2*-targeting siRNA (B), unirradiated or irradiated with 8-Gy x-ray 30 min before harvesting, was precipitated using the anti-ATM antibody or normal rabbit IgG and visualized by Western blotting using the anti-phospho-ATM (Ser 1981) antibody and the anti-ATM antibody. 30 μg of each lysate was also subjected to Western blot analysis using the anti-FLAG-antibody and anti-CDK2 antibody. Please note that the membranes were not reprobed because the band of ATM is not distinguishable from that of phosphorylated ATM. **(C)** Immunofluorescence visualization of foci of ATM phosphorylated on Ser 1981 (green) in MCF7 cells transfected with nontargeting control siRNA (upper panel) and in MCF7 cells transfected with *SYCE2*-targeting siRNA (lower panel), stained at 10 min after 0.2-Gy x-ray irradiation. Scale bar, 20 μm. **(D, E)** Percentages of cells containing three or more large phospho-ATM (Ser 1981) foci. Columns and bars represent the mean of three independent experiments and SD, respectively. A total of 100 cells were examined for each cell line. **(F, G)** Sensitivities of the mock cells and FLAG-SYCE2–expressing RPE cells treated with either DMSO alone or 10 μM KU-55933 dissolved in DMSO for 24 h before x-ray irradiation (F) and 1 h treatment with cisplatin (G).

*FLAG-SYCE2* construct. Conversely, stable FLAG-SYCE2 expression increased the frequency of radiation-induced foci formation of phosphorylated ATM (Ser 1981) from 42.5 ± 3.3% (mean ± SD) in mock cells to 65.0 ± 10.3% and 58.5 ± 6.2% in the two FLAG-SYCE2–expressing RPE clones (Fig 3E). Of note, the frequencies of foci formation of phosphorylated ATM (Ser 1981) in the steady state also increased from 0.3 ± 0.6% in mock cells to 8.2 ± 1.3% and 8.3 ± 2.6% in the two FLAG-SYCE2–expressing clones, indicating that SYCE2 potentiates the ATM activity even in the absence of exogenous DNA damage.

### Inhibition of ATM activity abrogates the phenotypes observed in SYCE2-expressing cells

We next investigated the roles of ATM in inducing the resistance to DNA-damaging agents observed in SYCE2-expressing cells. We inhibited ATM activity by treating the mock cells and FLAG-SYCE2–expressing RPE cells with an ATM-specific inhibitor, KU-55933 (Hickson et al, 2004), at a concentration of 10 μM and then observed their sensitivities to x-ray irradiation and cisplatin. Although the treatment with the ATM inhibitor increased the sensitivities to x-ray irradiation and cisplatin both in the mock cells and SYCE2-expressing cells, it abrogated the resistance to these DNA-damaging agents observed in SYCE2-expressing cells, which showed sensitivities comparable with those in the mock cells (Fig 3F and G). This result indicates that increased ATM activation contributes to the induction of resistance to DNA-damaging agents in SYCE2-expressing cells.

We also confirmed the roles of ATM in promoting DSB repair pathways in SYCE2-expressing MCF7 cells (Fig S3C and D). We inhibited ATM activity by treating the cells with 10 μM KU-55933 for 24 h and then analyzed the radiation-induced foci formation of 53BP1 and RAD51. Although the treatment with KU-55933 reduced the frequencies of 53BP1 and RAD51 foci-positive cells in both control MCF7 cells and the MCF7 cells treated with *SYCE2*-targeting siRNA, it induced an especially large reduction in the frequency of 53BP1 foci-positive cells and RAD51 foci-positive cells in control cells from 84.3 ± 6.0% to 51.3 ± 8.3% and 44.3 ± 7.1% to 5.3 ± 2.3%, respectively, which levels are comparable with the corresponding frequencies observed in the MCF7 cells treated with *SYCE2*-targeting siRNA. These results indicate that increased ATM activity in SYCE2-expressing cells contributes to the promotion of both NHEJ and HR.

On the other hand, SYCE2 expression affected neither the cell-cycle distribution nor the levels of phosphorylation of Chk2 on Thr 68 and phosphorylation of p53 on Ser 15 (Figs S2C and S3E), indicating that ATM activation in SYCE2-expressing cells affects ATM-dependent DSB repair without affecting the cell-cycle checkpoints.

### SYCE2 expression inhibits the domain formation of HP1α and reduces the colocalization of H3K9me3 and HP1α

We next tried to identify the molecules that are directly targeted by SYCE2, searching for the factors that could affect the steady-state ATM activity. Immunofluorescence of the cells stably expressing FLAG-SYCE2 revealed that the SYCE2 protein is localized diffusely in the nucleus and in the cytoplasm (Fig 4A). Because the nuclear architecture and chromatin environment were recently reported to

regulate the DNA damage response and repair, we next investigated whether the distribution of H3K9me3, a heterochromatin marker, is affected by SYCE2 expression. Immunofluorescence showed colocalization of the H3K9me3 domains with the regions densely stained with DAPI in RPE cells (Fig S4A), whereas heterochromatin domains were not clearly identified by DAPI staining in MCF7 cells (Fig 4B). The distributions of H3K9me3, namely heterochromatin, were not affected by the silencing of SYCE2 in MCF7 cells or the stable expression of FLAG-SYCE2 in RPE cells (Figs 4B and S4A). We next investigated whether the distribution of another heterochromatin-related protein HP1α is affected by SYCE2 expression. Immunofluorescence images of large fields stained with anti-HP1α antibody suggest that MCF7 cells are heterogeneous, containing both cells with large HP1α domains and others without (Fig S4B). Surprisingly, the percentages of the cells containing large HP1α domains in the nucleus became apparent after knocking down SYCE2 in MCF7 cells (Figs 4C and S4B). The silencing of SYCE2 in MCF7 cells increased the domain formation of HP1α from 42.5 ± 4.9% to 82.7 ± 5.7% in the steady state and from 36.6 ± 8.4% to 83.1 ± 4.5% upon 8-Gy x-ray irradiation (Fig 4D). Importantly, this effect of SYCE2 on HP1α localization was also observed in the non-cancerous HEK293 cells, which show very low *SYCE2* expression compared with MCF7 cells (Fig S1A). The HP1α domains became apparent after knocking down SYCE2 in HEK293 cells (Fig S4C and D). The silencing of SYCE2 in HEK293 cells also increased the domain formation of HP1α from 8.3 ± 2.5% to 25.3 ± 1.2% in the steady state (Fig S4E). These results suggest that the effect of SYCE2 on HP1α localization could be observed regardless of its expression levels.

Then, what are the large HP1α domains observed in these figures? Are they heterochromatin (H3K9me3 domains) or the sites of DNA repair? To sort this out, we first carried out a double staining immunofluorescence study using antibodies for HP1α and H3K9me3. The silencing of SYCE2 in MCF7 cells induced colocalization of the HP1α domains and H3K9me3 (Fig S4F). Conversely, FLAG-SYCE2–expressing RPE cells showed reduced colocalization of the HP1α domains and H3K9me3 from 56.0 ± 8.1% to 20.0 ± 6.5% in the steady state (Fig 4E and F). The expression levels of H3K9me3 and HP1α were not affected by the expression levels of SYCE2 (Fig S4G and H). We next performed double staining immunofluorescence studies using antibodies for HP1α and γH2AX to test whether the HP1α domains are colocalized with the γH2AX foci induced after 1-Gy irradiation (Fig S4I). The percentages of HP1α domains colocalized with γH2AX were very small; 5.3 ± 0.58 % in MCF7 cells treated with nontargeting control siRNA and 4.3 ± 0.58 % in MCF7 cells treated with siRNA targeting *SYCE2*, showing no significant differences between these two types of cells (*P* > 0.05). Taken together, these results indicate that the large HP1α domains are located at heterochromatin (H3K9me3 domains) rather than the sites of DNA repair. Thus, the fact that SYCE2 expression inhibits the domain formation of HP1α suggests that SYCE2 might dissociate HP1α from heterochromatin.

### The N-terminal hydrophobic amino acids of SYCE2 directly bind to the chromoshadow domain of HP1α

Because SYCE2 expression dynamically changed the distribution of HP1α as described above, we hypothesized that HP1α might be the direct target of SYCE2. We therefore tested whether SYCE2 interacts

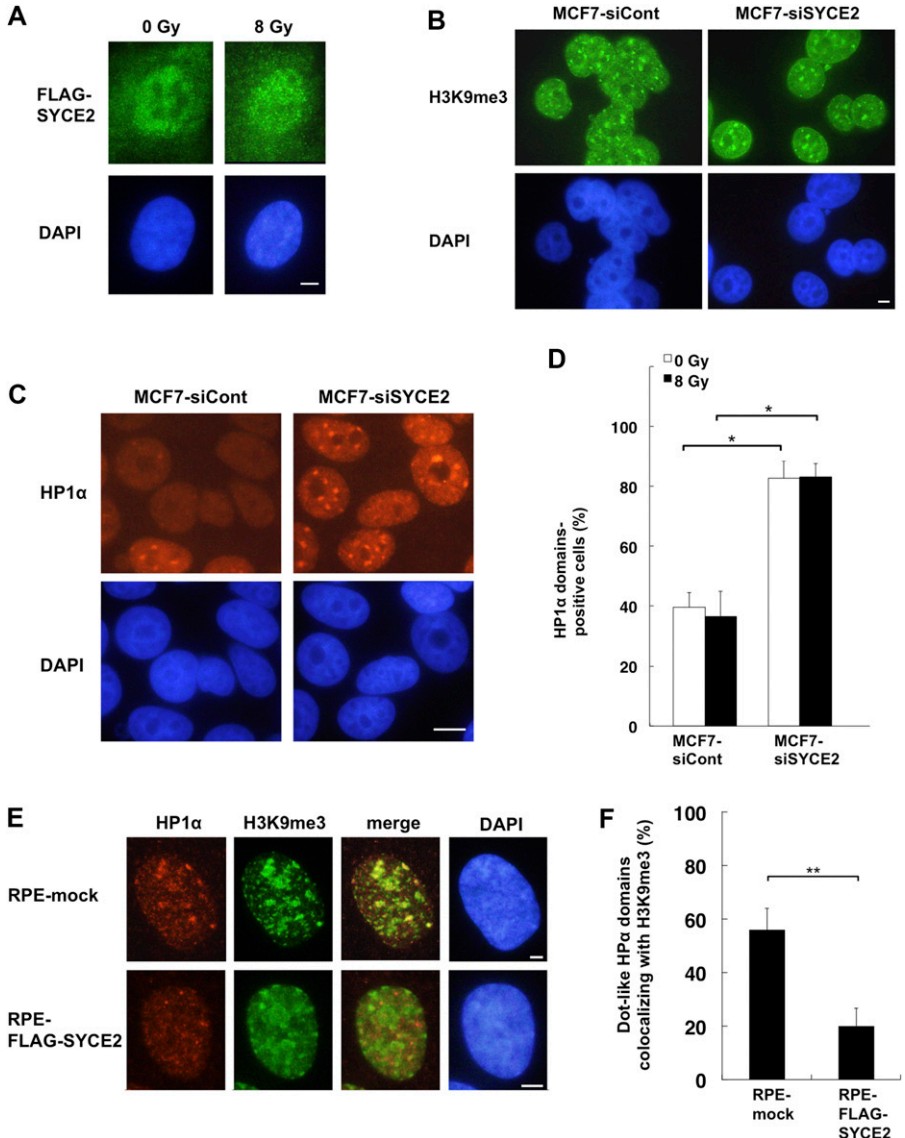

**Figure 4.   SYCE2 expression changes the localization of HP1α in the nucleus without affecting the localization of H3K9me3.**
**(A)** Immunofluorescence visualization of the SYCE2 protein in RPE cells stably expressing FLAG-SYCE2, unirradiated or x-irradiated with 8 Gy. Cells were stained with the anti-FLAG antibody (green) and DAPI (blue) before irradiation or 1 h after irradiation. Scale bar, 10 μm. **(B)** Immunofluorescence visualization of the H3K9me3 protein in unirradiated MCF7 cells transfected with nontargeting control siRNA or *SYCE2*-targeting siRNA. Cells were stained with the anti-H3K9me3 antibody (green) and DAPI (blue). Scale bar, 10 μm. **(C)** Immunofluorescence visualization of the HP1α protein in unirradiated MCF7 cells transfected with control siRNA or *SYCE2*-targeting siRNA. Cells were stained with the anti-HP1α antibody (red) and DAPI (blue). Scale bar, 10 μm. **(D)** Percentages of cells containing large HP1α domains. Columns and bars represent the mean of three independent experiments and SD, respectively. A total of 100 cells were examined for each cell type. **(E)** Immunofluorescence visualization of HP1α and H3K9me3. Cells were stained with anti-HP1α (red), anti-H3K9me3 (green) antibodies, and DAPI (blue). In the "merge" panel, the red panel and green panel are merged to evaluate the colocalization of HP1α and H3K9me3. Scale bar, 10 μm. **(F)** Percentages of HP1α domains colocalizing with H3K9me3 are indicated. Columns and bars represent the mean of three independent experiments and SD, respectively. A total of 100 HP1α domains were examined for each cell type.

with HP1α. The anti-FLAG antibody pulled down HP1α in cells expressing FLAG-SYCE2 (Fig 5A), indicating that SYCE2 forms a complex with HP1α. The interaction of SYCE2 and HP1α did not change after x-ray irradiation (Fig S5A). To further define the regions of HP1α and SYCE2 that interact with each other, we constructed deletion mutants of SYCE2 tagged with FLAG and deletion mutants of HP1α tagged with HA epitopes (Fig 5B), and examined whether they can interact with each other when they are transiently co-transfected in COS7 cells. The full-length FLAG-SYCE2 protein was found to bind to both the deletion mutant lacking the chromodomain of HP1α and the deletion mutant lacking the chromodomain and the intervening region of HP1α (Fig 5C). Because both of the mutants contain the chromoshadow domain, this result suggests that SYCE2 may interact with the chromoshadow domain of HP1α. In support of this notion, the full-length FLAG-SYCE2 protein did not bind to the deletion mutant lacking the chromoshadow domain of HP1α (Fig 5D). Then, we checked the region of SYCE2 that

interacts with the chromoshadow domain of HP1α. The deletion mutant of HP1α containing only the chromoshadow domain interacted with the N-terminal region of SYCE2 (amino acids 1–88) (Fig 5E), but could not bind to the deletion mutant of SYCE2 lacking the amino acids 1–56 or 1–87 (Fig 5F), indicating that the N-terminal region (amino acids 1–56) of SYCE2 is required for interaction with the chromoshadow domain of HP1α.

This N-terminal region of SYCE2 (amino acids 1–56) contains a sequence of PHVKC (amino acids 9–13), which has a similarity to the canonical PXVXL pentapeptide motifs (Fig 5G), recognized for the binding with the chromoshadow domain of HP1 (Thiru et al, 2004). Moreover, the N-terminal region of SYCE2 (amino acids 1–56) also contains a hydrophobic sequence of LTVLEGKS (amino acids 49–56) (Fig 5G), which could also contribute to the interaction with the chromoshadow domain as is observed in other cases (Mendez et al, 2011, 2013). We therefore constructed another deletion mutant of SYCE2 lacking both the PHVKC and LTVLEGKS motifs and examined

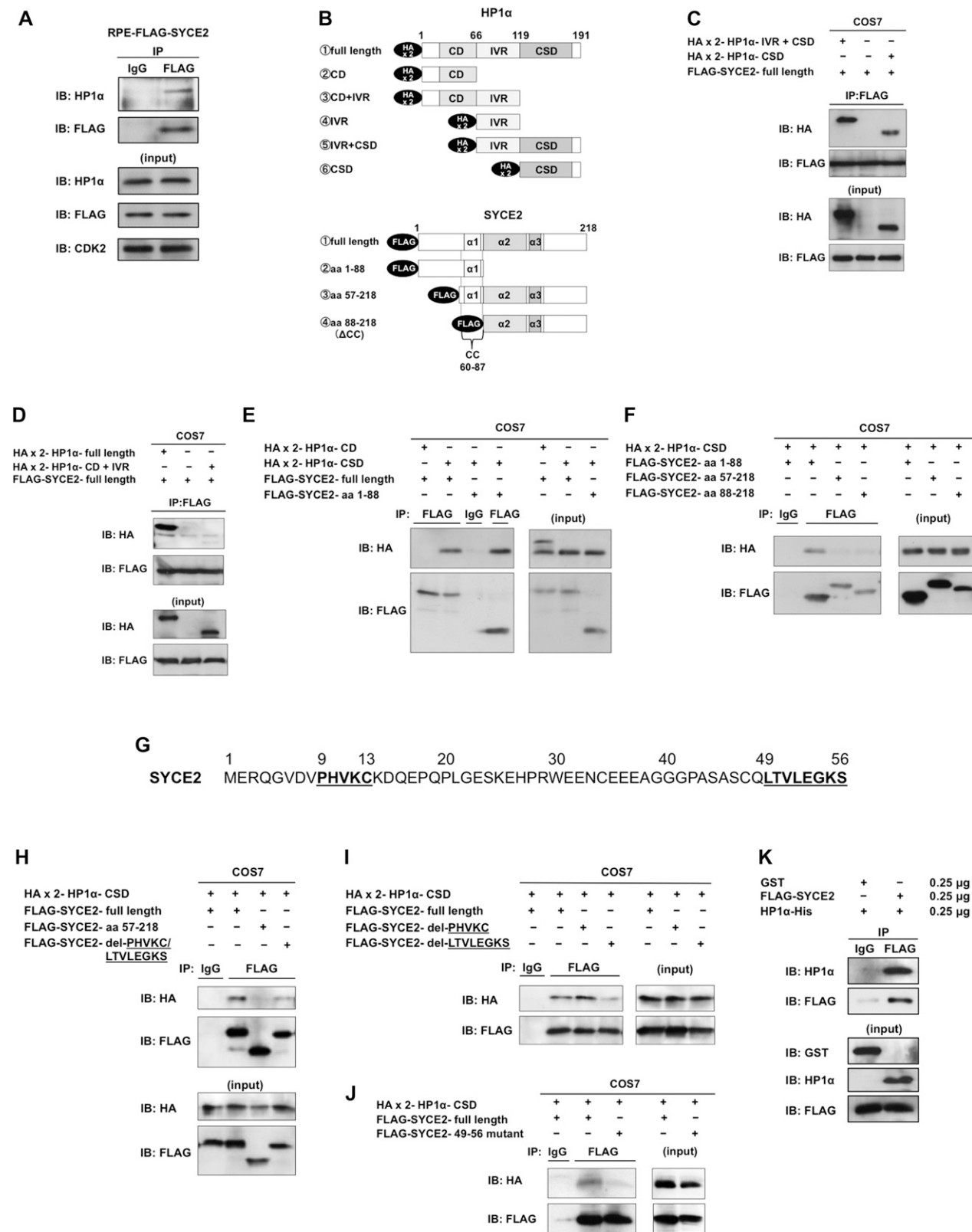

**Figure 5. SYCE2 directly binds to the chromoshadow domain of HP1α.**

**(A)** Interaction of FLAG-SYCE2 with HP1α in FLAG-SYCE2-expressing RPE cells. 500 μg of total cell lysates was immunoprecipitated using the anti-FLAG antibody produced in mouse or normal mouse IgG as a negative control and visualized by Western blotting using the anti-HP1α antibody produced in goat as a primary antibody and an HRP-linked anti-goat IgG antibody produced in rabbit as a secondary antibody. **(B)** Schematic diagrams of the HA-tagged deletion mutants of HP1α and the

whether this mutant can interact with the chromoshadow domain of HP1α. Deletion of both of these motifs resulted in a remarkable reduction in the interaction with the chromoshadow domain (Fig 5H), suggesting that either of these motif(s) may be critical for binding with the chromoshadow domain. We next examined whether the deletion mutants of SYCE2 lacking only the PHVKC or the LTVLEGKS sequence can interact with the chromoshadow domain of HP1α. Unexpectedly, the mutant of SYCE2 lacking only the PHVKC motif could interact with the chromoshadow domain at a level similar to that of the full-length SYCE2 protein, whereas the mutant of SYCE2 lacking only the hydrophobic LTVLEGKS sequence showed a remarkable reduction in the interaction with the chromoshadow domain (Fig 5I). We also constructed a "FLAG-SYCE2-49-56 mutant" harboring point mutations resulting in the substitution of the hydrophobic LTVLEGKS sequence into a non-hydrophobic ETDEEENS sequence. The "FLAG-SYCE2-49-56 mutant" also showed a remarkable reduction in the interaction with the chromoshadow domain (Fig 5J). These results indicate that the hydrophobic LTVLEGKS sequence is critical for the interaction with the chromoshadow domain of HP1α.

We next examined whether SYCE2 directly binds to HP1α. The recombinant protein for SYCE2 pulled down the recombinant protein HP1α (Fig 5K), indicating that SYCE2 directly binds to HP1α.

### Expression of SYCE2 inhibits the interaction of HP1α with H3K9me3

HP1α binds to H3K9me3 through its chromodomain and contributes to heterochromatin formation (Bannister et al, 2001; Lachner et al, 2001). We therefore tested whether the interaction between HP1α and H3K9me3 is affected by SYCE2 expression. Compared with mock cells, the FLAG-SYCE2–expressing RPE cells showed a reduction in the binding of HP1α and H3K9me3, whereas the protein levels of HP1α were not affected (Fig 6A). Conversely, the interaction of these two proteins was recovered by silencing SYCE2 in MCF7 cells (Fig 6B), indicating that the expression of SYCE2 inhibits the interaction between HP1α and H3K9me3.

We next tested whether the recombinant protein of SYCE2 can inhibit the direct binding between the recombinant proteins HP1α and H3K9me3. Addition of SYCE2 inhibited the direct binding of HP1α and H3K9me3 in a dose-dependent manner (Fig 6C). Taken together, the results indicate that SYCE2 inhibits the direct interaction between HP1α and H3K9me3 through its direct binding to the chromoshadow domain.

We further performed a salt-extractability assay in mock cells and RPE cells stably expressing FLAG-SYCE2 to test whether SYCE2 makes HP1α and H3K9me3 (or histone H3) extracted in different fractions. As previously described (Shechter et al, 2007), salt extraction is expected to differentially extract proteins according to their binding affinities to chromatin or nucleosome. In this assay, we extracted the nuclear fraction in three steps by increasing the concentrations of NaCl in the extraction buffer (0.5 M NaCl, 1.5 M NaCl, and 2.5 M NaCl) (Fig S6A). As expected, the H3K9me3 proteins were mostly extracted at a higher salt concentration of 1.5 M NaCl in the S2 fraction in mock cells, confirming that most parts of H3K9me3 would be tightly incorporated into chromatin (Fig S6B). On the other hand, histones H3 were mainly extracted at 1.5 M NaCl (in the same S2 fraction as H3K9me3), but a smaller pool of the histones H3 was also extracted at 0.5 M NaCl (in the S1 fraction) in mock cells, which may reflect the existence of free histones, either newly synthesized or ejected from chromatin, that are not incorporated into chromatin. The extraction patterns of H3K9me3 and histone H3 were similar in mock cells and in RPE cells expressing FLAG-SYCE2, which may support the findings that SYCE2 expression does not affect the distributions of H3K9me3 (Figs 4B and S4A). Unexpectedly, HP1α proteins were mainly extracted at a lower salt concentration of 0.5 M NaCl in the S1 fraction and only a small pool of HP1α proteins was extracted at 1.5 M NaCl (in the same S2 fraction as H3K9me3) even in mock cells. This suggests that the binding of HP1α to H3K9me3 to form heterochromatin may be loose and that most parts of the HP1α proteins are not tightly incorporated into chromatin even in mock cells showing very low expression of SYCE2. Under such situations, no differences in the extraction patterns of HP1α were detectable in the mock cells and RPE cells expressing FLAG-SYCE2, indicating that dissociation of HP1α from heterochromatin is hardly detectable by this assay. FLAG-SYCE2 in RPE cells was extracted at 0.5 M NaCl in the S1 fraction, as expected from its diffuse localization in the nucleus (Fig 4A). To conclude the results of the salt-extractability assay, we can at least say that SYCE2 does not affect the salt-extractability of either H3K9me3 at the nucleosome or total histone H3 including free histone H3.

### Dissociation of HP1α from H3K9me3 is critical to potentiation of steady-state ATM activity

We next examined whether dissociation of HP1α from H3K9me3 can potentiate steady-state ATM activity, as is observed in SYCE2-expressing cells. We first knocked down the endogenous HP1α in MCF7 cells, and then exogenously expressed the HA-tagged wild-type HP1α protein or the HA-tagged mutants of HP1α, which contained mutations in the conserved residues in the HP1α chromodomain (V23M and KW41/42AA) without affecting H3K9me3 protein levels (Fig S7A). Both of the mutants have previously been reported to abolish the binding ability of HP1α to H3K9me3 (Bannister et al, 2001; Lachner et al, 2001). Consistent with these previous reports, a pull-down assay revealed that the MCF7 cells expressing HA-tagged mutants of HP1α (V23M and KW41/42AA) cannot bind to H3K9me3 (Fig 7A).

FLAG-tagged SYCE2. **(C–F, H–J)** Analyses of interactions between each deletion mutant (C–F, H, I) or the mutant causing amino acid substitutions (J) of SYCE2 tagged with FLAG and that of HP1α tagged with HA. The "FLAG-SYCE2-49-56 mutant" in (J) has point mutations resulting in the substitution of the hydrophobic LTVLEGKS sequence in the amino acids 49–56 of SYCE2 into a non-hydrophobic ETDEEENS sequence. Expression vectors for the indicated mutants of SYCE2 and HP1α were transiently co-transfected into COS7 cells. 500 μg of total cell lysates of these cells was immunoprecipitated with the anti-FLAG antibody or normal IgG and then analyzed by immunoblotting with the anti-HA antibody or the anti-FLAG antibody. 30 μg of the input was also analyzed by immunoblotting using the anti-HA or anti-FLAG antibody. **(G)** Sequence of the N-terminal region (amino acids 1–56) of the SYCE2 protein. The PHVKC sequence similar to the canonical PXVXL motif and the hydrophobic sequence LTVLEGKS are underlined. **(K)** Pull-down assay using the indicated amounts of recombinant proteins for FLAG-tagged full-length SYCE2 in the presence of GST as a negative control experiment (left lane) and full-length HP1α (right lane). CD, chromodomain; IVR, intervening region; CSD, chromoshadow domain; α1-3, α-helical structures; CC, coiled-coil domain.

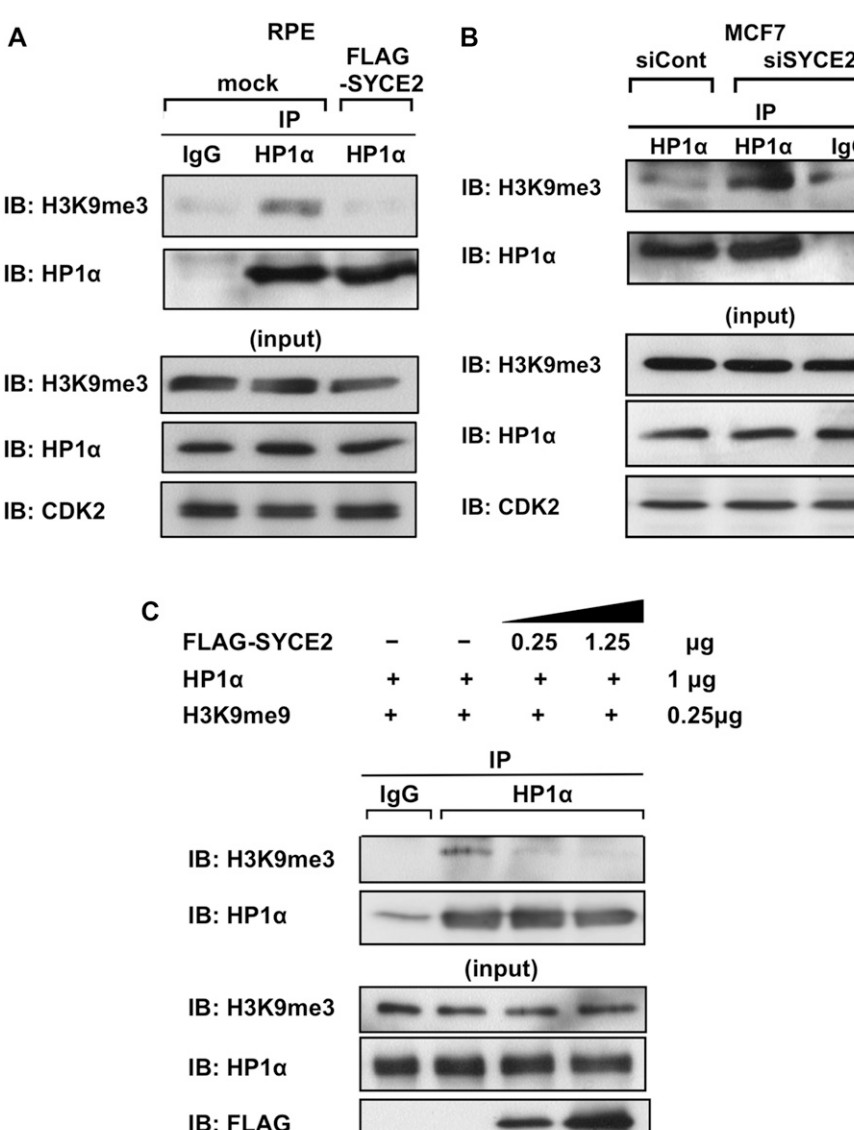

**Figure 6. Expression of SYCE2 inhibits the interaction of HP1α with H3K9me3.**
**(A, B)** Coimmunoprecipitation of HP1α and H3K9me3 from lysates of mock cells and RPE cells stably expressing FLAG-SYCE2 (A) or from MCF7 cells transfected with a nontargeting siRNA control or *SYCE2*-targeting siRNA (B). Protein complexes were immunoprecipitated using the anti-HP1α antibody or normal goat IgG as a negative control and visualized by Western blotting using the anti-H3K9me3 or the anti-HP1α antibody. **(C)** Protein–protein binding assay using recombinant proteins for FLAG-SYCE2, HP1α, and H3K9me3. The HP1α recombinant protein was incubated with the indicated amounts of the FLAG-tagged SYCE2 recombinant protein for 30 min, followed by addition of the indicated amounts of the H3K9me3 recombinant protein for 1 more hour. The H3K9me3 protein bound to HP1α was detected by immunoprecipitations using the anti-HP1α antibody or normal IgG as a negative control and subsequent Western blot analysis using the anti-H3K9me3 antibody.

We next performed immunofluorescence analysis to assess whether these mutants of HP1α lacking the ability to bind H3K9me3 could affect the autophosphorylation levels of ATM (Fig 7B). After 0.2-Gy irradiation, the frequency of cells with three or more phosphorylated ATM (Ser 1981) foci was 3.7 ± 1.5% (mean ± SD) in MCF7 cells expressing the exogenous wild-type HP1α protein, whereas those in MCF7 cells expressing HP1α V23M and KW41/42AA significantly increased to 13.3 ± 2.1% and 9.7 ± 2.0%, respectively. Moreover, the frequencies of foci-positive cells for phosphorylated ATM in the absence of exogenous DNA damage were also significantly increased to 2.3 ± 0.58% and 2.2 ± 0.29% in MCF7 cells expressing exogenous mutants of HP1α, V23M, and KW41/42AA, respectively, compared with 0.83 ± 0.29% in MCF7 cells expressing the exogenous wild-type HP1α protein. This result indicates that mutants of HP1α lacking the ability to bind H3K9me3 increase the ATM activity even in the absence of exogenous DNA damage. As our finding that the hydrophobic LTVLEGKS sequence is critical for the

binding with the chromodomain of HP1α protein suggests that this sequence may affect the ATM activity by dissociating HP1α from H3K9me3, we next constructed RPE cells stably expressing the FLAG-SYCE2 protein lacking this hydrophobic sequence (Fig S7B). We confirmed by immunofluorescence using an anti-FLAG antibody that this mutant is located mostly in the nucleus, retaining the same subcellular localization as the wild-type SYCE2 protein (Figs 4A and S7C). Although stable expression of the full-length FLAG-SYCE2 protein increased the frequency of foci formation of phosphorylated ATM (Ser 1981) both in the absence and in the presence of exogenous DNA damage as described in Fig 3E, the RPE cells expressing the mutant SYCE2 lacking the hydrophobic LTVLEGKS sequence showed no significant changes in the frequencies of cells positive for autophosphorylated ATM foci compared with mock cells (Fig 7C). Thus, the dissociation of HP1α from H3K9me3 observed in SYCE2-expressing cells might potentiate the steady-state ATM activity. In SYCE2-expressing MCF7 cells, the foci of phosphorylated

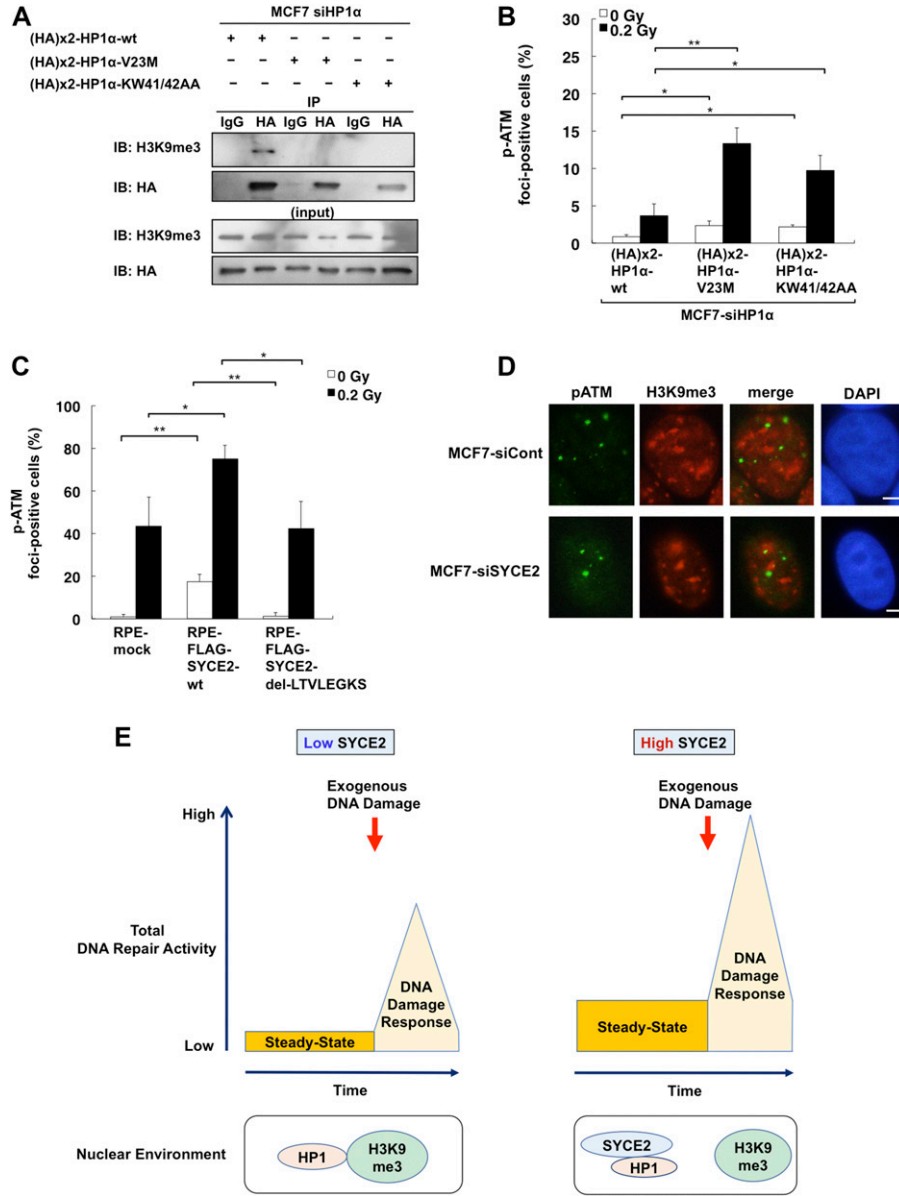

**Figure 7. Dissociation of HP1α from H3K9me3 is critical to potentiation of steady-state ATM activity.**
**(A)** Immunoprecipitation and Western blot analysis showing that mutants of HP1α (V23M and KW41/42AA) cannot bind H3K9me3. After knocking down the endogenous HP1α protein, the HA-tagged wild-type HP1α protein or the HA-tagged mutants of HP1α (V23M or KW41/42AA) were exogenously expressed in MCF7 cells. 500 μg of the lysates from these cells was immunoprecipitated with either normal IgG or anti-HP1α antibody and then analyzed by immunoblotting using the anti-H3K9me3 antibody or the anti-HA antibody. **(B, C)** Percentages of cells containing three or more large phospho-ATM (Ser 1981) foci. Columns and bars represent the mean of three independent experiments and SD, respectively. A total of 100 cells were examined for each cell type. **(D)** Immunofluorescence visualization of ATM phosphorylated on Ser 1981 and H3K9me3. Cells were stained with anti-phospho ATM (Ser 1981) (green) and anti-H3K9me3 (red) antibodies and DAPI (blue). In the "merge" panel, the red panel and green panel are merged to evaluate the localization of phospho-ATM (Ser 1981) (green) and H3K9me3 (red) in a single cell. Scale bar, 10 μm. **(E)** Schematic representation showing that expression levels of SYCE2 in somatic cells define the steady-state ATM activity by affecting HP1 localization. When the expression level of SYCE2 is low (left), HP1 is bound to H3K9me3, and the steady-state ATM activity is kept low. On the other hand, when the expression level of SYCE2 is high (right), direct binding of SYCE2 to HP1 insulates HP1 from H3K9me3, namely heterochromatin. As a result, HP1 will be distributed in euchromatin as well, which will potentiate the steady-state ATM activity and increase the levels of total DNA repair activity when DNA damage is induced exogenously.

ATM (Ser 1981) were mostly not colocalized with H3K9me3 or regions densely stained with DAPI, and were not affected by the knockdown of SYCE2 (Fig 7D). This result indicates that ATM is mainly activated in euchromatin, which is not affected by the expression levels of SYCE2.

## Discussion

We identified a novel partnership between the synaptonemal complex protein SYCE2 and HP1α, showing that SYCE2 directly binds to HP1α and dissociates HP1α from H3K9me3, which leads to potentiation of the steady-state ATM-dependent DNA repair activity and increases the total DNA repair activity upon the induction of exogenous DNA damage (Fig 7E). The changes in the HP1α localization

were observed in both non-cancerous cells expressing low levels of SYCE2 and cancer cells expressing rather high levels of SYCE2. Thus, our findings suggest that the varying levels of SYCE2 expression might impact the cellular ability to respond to DNA damage in a way that seems to depend on altered HP1α chromatin binding and changes in ATM activity. Our findings shed light on a hitherto unappreciated function of SYCE2 in somatic cells with potential relevance for cancer. As SYCE2 is expressed at varying elevated levels in cancer cells as opposed to normal cells, SYCE2 might be a prime candidate target for cancer selective therapy. Although many attempts have been made to identify novel HP1-binding factors, SYCE2 has not yet been identified because the expression levels of SYCE2 in somatic cells are lower than those in the germ line.

SYCE2 is a unique HP1α-associated protein whose chromoshadow domain-mediated interaction dissociates HP1α from H3K9me3. This

function is in contrast to that of the recently reported HP1-associated protein SENP7, where the chromoshadow domain-mediated interaction with HP1 increases the binding capability between HP1α and H3K9me3, and formation of large HP1α domains was disrupted by silencing SENP7 (Maison et al, 2012; Romeo et al, 2015). This difference might be explained by the manner how SYCE2 recognizes the chromoshadow domain of HP1α. SYCE2 contains a sequence of PHVKC similar to the canonical PXVXL pentapeptide motifs, which are shown to be responsible for HP1 binding in other proteins including not only SENP7 but also Kruppel-associated box domain–associated protein 1, Shugoshin, Su(VAR)3-9, and ACF1 (Lechner et al, 2000; Eskeland et al, 2007; Kang et al, 2011). However, in the current case, deletion of the PHVKC sequence of SYCE2 did not alter the binding ability to HP1α. Instead, the hydrophobic sequence of LTVLEGKS on the C-terminus side of the PHVKC sequence of SYCE2 was shown to be responsible for its binding with HP1α. It is likely that the interaction between the chromoshadow domain and the hydrophobic sequence of LTVLEGKS in SYCE2 is strong enough to dissociate HP1α from H3K9me3, highlighting the importance of hydrophobic sequences in the chromoshadow-mediated interactions. Although dissociation of HP1 from H3K9me3 was mostly caused by the phosphorylation of HP1 or related proteins in the previously reported cases (Hirota et al, 2005; Ayoub et al, 2008, 2009; Bolderson et al, 2012; Jang et al, 2014), our results demonstrate for the first time that dissociation of HP1 from H3K9me3 could be epigenetically induced by an exogenous insulator, SYCE2.

What is the exact mechanism that links dissociation of HP1α from H3K9me3 and potentiation of ATM activity in SYCE2-expressing cells? We showed that the previously reported HP1α mutants lacking the binding ability to H3K9me3 not only potentiated the steady-state ATM activity but also increased the DNA damage-induced ATM activity, suggesting the biological significance of dissociation of HP1α from H3K9me3 in activating the ATM-mediated DNA damage response. Moreover, we also showed that the cells expressing the mutant SYCE2 lacking the hydrophobic sequence of LTVLEGKS in SYCE2, which is critical to the dissociation of HP1α from H3K9me3, failed to show the potentiation of the steady-state ATM activity or increase in the DNA damage-induced ATM activity, indicating that the interaction between the chromoshadow domain of HP1α and the hydrophobic sequence of LTVLEGKS in SYCE2 may be critical in regulating ATM activity. Of note, the dissociation of HP1α from H3K9me3 in SYCE2-expressing cells did not necessarily indicate that the heterochromatin had changed into relaxed euchromatin because the distributions of H3K9me3 or the regions densely stained with DAPI were not affected by changes in the SYCE2 expression levels. It is likely that, in SYCE2-expressing cells, HP1α is dissociated from heterochromatin and widely located even in the euchromatin regions, which may play important roles in activating ATM and subsequent DSB repair. Indeed, as we have shown, the phosphorylated ATM was mostly located in euchromatin regions, regardless of the levels of SYCE2 expression. These observations are compatible with the recent proposal by Ayrapetov et al (2014) that HP1 may be recruited to DSBs in open chromatin domains with low H3K9me3 levels and form repressive chromatin by transiently increasing the levels of H3K9me3 and promoting efficient activation of ATM in these regions. It is also worth noting that that binding of SYCE2 to HP1α and the dissociation of HP1α

from H3K9me3 occurred even in the absence of induction of DSBs. Thus, SYCE2 may contribute to the generation of a nuclear environment in which HP1α is delocated from heterochromatin without affecting H3K9me3. Such a nuclear environment could potentiate the steady-state ATM-mediated DNA repair and increase the total DNA repair activity upon the induction of exogenous DNA damage, which would finally cause radioresistance and cisplatin resistance of the SYCE2-expressing cells.

ATM autophosphorylation has been observed in the early stages of cancers of the breast, colon, lung, skin, testes, and urinary bladder, suggesting that the DNA damage response pathways are constitutively hyperactivated during the formation of human cancers (Bartkova et al, 2005a, b; Gorgoulis et al, 2005). Although these findings suggest that the DNA damage response might serve as an anti-cancer barrier to prevent early stages of tumorigenesis, a recent report showed that the ATM kinase is also hyperactivated in the late stages of breast tumor tissues with lymph-node metastasis, providing evidence that the DNA damage response may also play a role in the late stages of tumor progression and metastasis (Sun et al, 2012). This increased damage response can be an excellent target of cancer therapy (Hosoya & Miyagawa, 2014).

What is the mechanism for the constitutive activation of ATM in cancer? Recent reports have suggested the importance of chromatin alterations in activating ATM-dependent signaling even in the absence of exogenous DNA damage. For example, a recent report showed that activation of the ATM-mediated DNA damage response does not require exogenous DNA damage and that stable association of repair factors with chromatin may be the critical step in triggering, amplifying, and maintaining the DNA damage response signal (Soutoglou & Misteli, 2008). Other reports showed that the nucleosome-binding protein HMGN1 promotes ATM activation by regulating the steady-state intranuclear interaction of ATM with chromatin before the induction of DNA damage, and predetermines the kinetics of ATM activation after the induction of exogenous DNA damage (Kim et al, 2009; Gerlitz & Bustin, 2009). More recently, chromatin alterations induced by histone hyperacetylation or HP1 depletion were shown to activate the ATM-dependent signaling (Kaidi & Jackson, 2013). Although these studies link chromatin alterations to ATM activation, most of them were performed under non-physiological conditions.

The dissociation of HP1α from H3K9me3 by SYCE2 shown in our study might be one of the mechanisms for the increased steady-state ATM activity in cancer. Our results clearly demonstrate that changes in HP1α localization have a great impact on the ATM activity, both in the steady state and in the presence of exogenous DNA damage. Interestingly, among the numerous downstream effectors of ATM, only the functions of DNA repair proteins were affected by SYCE2, whereas the phosphorylation of cell cycle checkpoint proteins was not affected. Accumulating evidence suggests that ATM plays diverse roles in the DNA damage response and in the responses to various stimuli or physiological situations, by operating in different signalling pathways (Shiloh & Ziv, 2013). It may be possible that ATM selects pathways to transmit the signals corresponding to the modes of ATM activation. Thus, the SYCE2 expression levels, which vary among cancers, may define the background ATM-mediated DNA repair capacity of each cancer.

Proliferating cancer cells are characterized by increased rates of DNA replication, indicating that cancer cells suffer from increased levels of replication stress. This is likely to cause DSBs resulting from stalled replication forks, which can be mainly repaired by HR. This mechanism may be explained by our finding that the ATM-mediated HR repair activity was increased in SYCE2-expressing cancer cells.

# Materials and Methods

### Key resources

The sources and catalog numbers for key resources used in this study including antibodies, RNA samples, chemicals, recombinant proteins, and cell lines are listed in Table S1.

### Cell cultures

RPE cells were cultured in Dulbecco's modified Eagle's and Ham's F12 medium containing 0.25% additional sodium bicarbonate and 10% fetal bovine serum. MCF7 cells, HeLa-DRGFP cells, HEK293 cells, and COS7 cells were cultured in Dulbecco's modified Eagle's medium supplemented with 10% fetal bovine serum.

### Expression analysis of the cell lines by real-time RT–PCR

Total RNA from each cell line was extracted using RNA iso Plus (Takara Bio). 1 μg of total RNA was reverse-transcribed in a 20-μl reaction mixture with 2 units of PrimeScript reverse-transcriptase (Takara Bio) using a random hexamer (Takara Bio). Real-time RT–PCR was carried out with a Takara Smart Cycler II System (Takara Bio) in a 25-μl reaction volume containing 2 μl template cDNA using SYBR Premix Ex Taq II (Takara Bio) for the detection of PCR products. The PCR primer sets for *SYCE2* and *GAPDH* (primer set ID numbers HA122934 and HA067812, respectively) were purchased from Takara Bio. The expression level of the *SYCE2* gene was evaluated as the ratio of its mRNA to that of *GAPDH* mRNA.

### Expression analysis using the tumor/normal tissues by conventional RT–PCR

5 μl of cDNA was analyzed for *SYCE2* expression by PCR using primers SYCE2-F/SYCE2-R1. 1 μl of the synthesized cDNA was also analyzed for *GAPDH* expression by PCR using primers GAPDH-F/GAPDH-R (Table S2) as a control experiment. The sequences of all the RT–PCR products were verified by sequencing.

### Knockdown experiments

Transfection of the siRNA was performed using DharmaFECT 4 (Dharmacon). An siCONTROL nontargeting siRNA from Dharmacon was used as a negative control. The sequences for siRNA for *SYCE2* and *HP1α* are listed in Table S3. The siRNA targeting the 3' UTR of *SYCE2* or *HP1α* was used for rescue experiments expressing exogenous FLAG-SYCE2 or functional analyses of HP1α mutants

(V23M and KW41/42AA) under silencing of the endogenous HP1α protein.

### Exogenous expression of full-length *SYCE2* cDNA tagged with *FLAG* in RPE and MCF7 cells

Human *SYCE2* cDNA tagged with *FLAG* was amplified from human testis cDNA by PCR using FLAG-SYCE2-F/SYCE2-R2 primers (Table S2), followed by insertion into an expression vector containing the MMTV promoter (Hosoya et al, 2012). The vector construct was transfected into RPE cells using a Bio-Rad Gene Pulser at 1,200 V and 10 μF, followed by selection using 900 μg/ml Zeocin (Invitrogen). After 14 d, the colonies were isolated and expanded. The *FLAG*-tagged full-length *SYCE2* cDNA was also inserted into the pcDNA3.1-zeo expression vector (Invitrogen) and used in the rescue experiments with RNA interference and transfection in MCF7 cells.

### Immunoprecipitation

For proteins other than ATM and phospho-ATM (Ser 1981), the cells were lysed in NETN lysis buffer (150 mM NaCl, 1 mM EDTA, 20 mM Tris [pH 8.0], 0.5% Nonidet P-40, and 1 mM DTT) supplemented with 1 mM phenylmethylsulfonyl fluoride and 10 μg/ml aprotinin and were incubated in the presence of DNaseI at room temperature for 10 min. 500 μg of total cell lysates were incubated with the primary antibody for 1 h at 4°C, after which 15 μl of protein-A or protein-G agarose was added to each sample. After rotation for 1 h at 4°C, the immunoprecipitates were washed five times in NETN lysis buffer. To detect phospho-ATM (Ser 1981) and ATM, cell lysis, immunoprecipitation, and Western blot analysis were carried out as previously described (Sun et al, 2009). The cells were lysed in ATM lysis buffer. 500 μg of total cell lysates was incubated with the ATM antibody for 1 h at 4°C, after which 15 μl of protein-A agarose was added to each sample. After rotation for 1 h at 4°C, the immunoprecipitates were washed three times in ATM lysis buffer, and once each in high-salt buffer and base buffer. They were then subjected to Western blot analysis using the anti-phospho-ATM (Ser 1981) antibody (1:200 in dilution) or anti-ATM (Ab-3) antibody (1:50 in dilution) as a primary antibody.

### Western blot analysis

To detect proteins other than ATM and phospho-ATM (Ser 1981), the cells were lysed in NETN lysis buffer supplemented with 1 mM phenylmethylsulfonyl fluoride and 10 μg/ml aprotinin. Protein samples were subjected to SDS-polyacrylamide gel electrophoresis and electrotransferred onto Hybond-P membranes (GE Healthcare), and then reacted with primary antibodies and secondary antibodies at the appropriate dilutions. The blots were visualized by ECL Western blotting detection reagents (GE Healthcare).

### Cell survival assay

The cells in suspension were subjected to x-irradiation or incubation in the presence of cisplatin (Nihon-Kayaku) for 1 h and washed three times with PBS. The RPE cells were plated at a density of $10^3$ cells per 60-mm dish and grown for 7 d. The MCF7 cells were

plated at a density of 2 × 10$^3$ cells per 60-mm dish and grown for 14 d. After fixing and staining, colonies were counted. All measurements were performed in triplicate.

## Immunofluorescence

The cells were cultured on coverslips. They were either untreated or irradiated with 8 Gy. At 2 h after irradiation, the cells were fixed with 4% paraformaldehyde for 10 min at room temperature. The cells were then blocked with 10% horse serum and incubated with primary antibodies at 37°C for 1 h and with secondary antibodies at 37°C for 30 min. Finally, the cells were counterstained with DAPI and mounted. For the analyses of foci formation, a total of more than 100 cells per each cell type were counted in randomly selected fields using an Olympus BX51 fluorescence microscope. Images were captured with an Olympus DP80 digital camera using Olympus DP controller and DP manager softwares.

## In vivo DNA-ligation assay

To analyze the ability of the cells to join DNA DSB ends by the NHEJ pathway, an in vivo DNA-ligation assay was performed according to the methods described by Buck et al (2006) with slight modifications. Briefly, MCF7 cells transfected with either nontargeting siRNA or siRNA for *SYCE2* were transfected with the linearized pEGFP-C1 vector (Clontech) digested with the restriction enzymes KpnI and SacI (Takara Bio). DNA was extracted 48 h after transfection by using the Hirt extraction procedure (Guo et al, 2007), and real-time RT–PCR was carried out as described above to measure recircularized DNA and input DNA. The PCR primer sets for detecting the recircularized DNA were pEGFP-rejoining-F and pEGFP-rejoining-R, whereas those for detecting input DNA were pEGFP-input-F and pEGFP-input-R (Table S4). The proportions of recircularized DNA were calculated as the ratio to input linearized DNA. The junctions of the recircularized plasmids were also analyzed by DNA sequencing.

## DR-GFP assay

To analyze the efficiency of the cells for repairing DSB by the HR pathway, a DR-GFP assay was performed using the previously described HeLa-DRGFP cells (Sakamoto et al, 2007). The non-targeting control siRNA or the siRNA for *SYCE2* was transfected into HeLa-DRGFP cells seeded in a 60-mm dish at 2 × 10$^5$ cells/dish using DharmaFECT4 (Dharmacon). 24 h later, 15 μg of the I-SceI expression vector was additionally transfected using Lipofectamine 2000 (Invitrogen). 48 h after transfection of the I-SceI expression vector, the percentage of GFP-positive cells was quantified using an EPICS XL flow cytometer (Beckman Coulter). The effect of knocking down *SYCE2* on the efficiency of HR was evaluated as the ratio of the GFP-positive cells in cells transfected with *SYCE2* siRNA to that in cells transfected with control siRNA.

## Cell cycle analysis

Cell cycle analysis was performed with an EPICS XL flow cytometer (Beckman Coulter) using a CycleTEST PLUS DNA Reagent kit (Beckton Dickinson).

## Vector constructions of the deletion mutants of SYCE2 and HP1α and transient expression in COS7 cells

Three types of the *FLAG*-tagged *SYCE2* mutant cDNA were amplified from the *FLAG*-tagged full-length *SYCE2* cDNA described above using the primer sets listed in Table S5. The *FLAG*-tagged *SYCE2*-aa 1–88 mutant cDNA was then cloned into the BamHI and XbaI sites of the pcDNA3.1-zeo expression vector (Invitrogen). The *SYCE2*-aa 57–218 and *SYCE2*-aa 88–218 mutants were cloned into the ClaI and XbaI sites of a modified pcDNA3.1-zeo expression vector containing the FLAG sequence. Six types of the *HP1α* fragment cDNA were amplified from the cDNA of MCF7 cells using the primer sets listed in Table S5, and were cloned into the ClaI and XbaI sites of the modified pcDNA3.1-zeo expression vector containing two repeats of HA sequences. All the constructs were sequenced to confirm the fidelity of the sequences and conservation of the reading frame. The expression vectors were transfected into COS7 cells using Lipofectamine 2000 Reagent (Invitrogen) according to the manufacturer's instructions. Cells cultured for 48 h after transfection were subjected to the next analyses.

## Protein–protein binding assay using recombinant proteins

The recombinant proteins for full-length SYCE2 tagged with FLAG, HP1α full-length protein, H3K9me3, and GST were obtained from OriGene, Abcam, Active Motif, and Sigma, respectively. To detect the direct binding of HP1α to SYCE2, the protein–protein binding assay was performed with HP1α or GST in the presence of the recombinant protein FLAG-tagged SYCE2 using a Magnetic DYKDDDDK Immunoprecipitation kit (Takara Bio). The immuno-precipitated proteins were subjected to Western blot analysis using antibodies against HP1α to detect their direct binding to SYCE2. To analyze the effect of SYCE2 on HP1α-H3K9me3 binding, 1 μg of the HP1α recombinant protein was first incubated with the indicated amounts of FLAG-tagged SYCE2 recombinant protein in 100 μl of binding buffer (20 mM Tris [pH 8.0], 150 mM NaCl, 1 mM EDTA, and 0.1% Triton X-100) for 30 min at room temperature. Then, 0.25 μg of the H3K9me3 recombinant protein was added to each reaction and incubated for one more hour at room temperature. Immunopre-cipitation was performed with 1 h rotation with normal IgG or the anti-HP1α antibody and one more hour of rotation with protein A agarose at 4°C, followed by five washings with washing buffer (50 mM Hepes [pH 7.5], 0.5 M NaCl, 1 mM EDTA, and 1% NP40). The proteins that copelleted with the agarose were denatured and analyzed by Western blot analysis.

## Mutagenesis

The *SYCE2* deletion mutants lacking one or both of the PHVKC sequence and the hydrophobic LTVLEGKS sequence, the "FLAG-SYCE2-49-56 mutant," which has point mutations resulting in the substitution of the hydrophobic LTVLEGKS sequence into a non-hydrophobic ETDEEENS sequence, and the *HP1α* mutants (V23M and KW41/42AA) were created by mutagenesis using a PrimeSTAR Mutagenesis Basal kit (Takara Bio). The primer sets used to create the mutants are listed in Table S6. The "FLAG-SYCE2-49-56 mutant" was created by repeating mutagenesis three times, by creating

a mutation of V51D first, then mutations of L49E and L52E, and finally mutations of G53E and K54N, using the primers indicated in Table S6.

## Salt-extractability assay

Salt-extractability assay was performed as described previously with some modifications (Shechter et al, 2007) (Fig S6A). The cells were washed once in PBS and incubated for 10 min, occasionally rotating, in 500 µl of PBS containing 0.2% NP-40 and 1 mM phenyl-methylsulfonyl fluoride and 10 µg/ml aprotinin. Aliquots of these whole-cell extracts (W) were removed and the remaining extracts were subjected to low-speed centrifugation (1,800 $g$, 10 min, 4°C), by which the soluble fraction and the insoluble nuclear fraction were separated. The pellet containing insoluble nuclear fraction was then suspended in 400 µl of 0.5 M NaCl-extraction buffer (0.5 M NaCl, 50 mM Tris–HCl [pH 8.0], and 0.05% NP-40). After vortexing in-termittently for 1 min (10 s on, 10 s off) and 30 min rotation at 4°C, the supernatant (S1) and pellet were separated by centrifugation (6,500 $g$, 10 min, 4°C). The pellet was then suspended in 400 µl of 1.5 M NaCl-extraction buffer (1.5 M NaCl, 50 mM Tris–HCl [pH 8.0], and 0.05% NP-40). After vortexing intermittently for 1 min (10 s on, 10 s off) and 30 min rotation at 4°C, the supernatant (S2) and pellet were sepa-rated by centrifugation (15,000 $g$, 10 min, 4°C). The pellet was finally suspended in 400 µl of 2.5 M NaCl-extraction buffer (2.5 M NaCl, 50 mM Tris–HCl [pH 8.0], and 0.05% NP-40). After vortexing intermittently for 1 min (10 s on, 10 s off) and 30 min rotation at 4°C, the supernatant (S3) and pellet (P) were separated by centrifugation (16,000 $g$, 10 min, 4°C). The pellet was suspended in 400 µl of a buffer containing 50 mM Tris–HCl (pH 8.0) and 2% SDS. Same volumes from each fraction (W, S1, S2, S3, and P) were subjected to Western blot analysis.

## Statistics

All data were derived from at least three independent experiments. Statistical significance was determined using two-tailed $t$ test. Significance was indicated as *$P$ < 0.05 and **$P$ < 0.01.

# Supplementary Information

# Acknowledgements

We are grateful to Maria Jasin for providing the DR-GFP construct. We also thank Kenshi Komatsu and Junya Kobayashi for providing us with the HeLa-DRGFP cells. This work was supported in part by JSPS KAKENHI grants JP23591836, JP25125705, JP26461881, JP16H01298, and JP17K10471 to N Hosoya and by grants from the Takeda Science Foundation and the Naito Foun-dation to N Hosoya.

## Author Contributions

N Hosoya: conceptualization, data curation, formal analysis, su-pervision, funding acquisition, validation, investigation, and writing—original draft, review, and editing.

M Ono: investigation.

K Miyagawa: conceptualization, data curation, formal analysis, supervision, and writing—review and editing.

## Conflict of Interest Statement

The authors declare that they have no conflict of interest.

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
