## [Reviewer comments · Life Science Alliance]

Somatic role of SYCE2: an insulator that dissociates HP1 α from H3K9me3 and potentiates DNA repair

Noriko Hosoya, Masato Ono, Kiyoshi Miyagawa
DOI: 10.26508/lsa.201800021

Review timeline:

Submission date:	16 January 2018
1 st Editorial Decision:	23 February 2018
1 st Revision Received:	12 May 2018
2 nd Editorial Decision:	1 June 2018
2 nd Revision Received:	3 June 2018
Accepted:	4 June 2018

Report:

(Note: Letters and reports are not edited. The original formatting of letters and referee reports may not be reflected in this compilation.)

1st Editorial Decision

23 February 2018

Thank you for submitting your manuscript entitled "Mitotic role of SYCE2: an insulator that dissociates HP1 α from H3K9me3 and potentiates DNA repair" to Life Science Alliance. The manuscript was assessed by expert reviewers, whose comments are appended to this letter. We invite you to submit a revision if you can address the reviewers' key concerns, as outlined here.

As you will see, both referees appreciate your work and provide constructive input on how to better support your data. We would thus like to invite you to revise your work along the lines suggested by the reviewers. Note that additional mutations in SYCE2 (referee #1, comment regarding Figure 5) are not needed for acceptance in Life Science Alliance.

Thank you for this interesting contribution to Life Science Alliance. We are looking forward to receiving your revised manuscript.

REFeree REPORTS

Reviewer #1 (Comments to the Authors (Required)):

In this manuscript, the authors have investigated the function of SYCE2 in somatic cells. This protein was initially described as a subunit of the synaptonemal complex formed between homologous chromosomes during meiotic prophase. Yet, mRNA from this gene is also detected in several tissue culture cells and in some tumors. The authors convincingly show that knock-down of this mRNA in MCF7 cells results in decreased resistance to DNA-damaging agents, increased number of foci of gammaH2Ax, decreased phosphorylation of ATM at S1981, decreased RAD51 foci formation, and decreased 53BP1 foci formation (but only in irradiated cells). The opposite was seen upon forced expression of FLAG-SYCE2 in RPE cells. The authors then provide some data suggesting that SYCE2 interferes with HP1 α foci formation by interacting with its dimerization (chromoshadow) domain. This interaction is not mediated by a classical PXVXL motif in SYCE2 but is abrogated by deletion of a hydrophobic domain of the protein. It is proposed that the HP1 α /SYCE2 interaction prevents HP1 α to interact with H3K9me3.

Evidence for the presence of SYCE2 transcript and protein outside the germ line is to be found on the Protein Atlas. Yet, this work seems to be the first to investigate the function of SYCE2 in somatic cells. The demonstration of a role for this protein in DDR is very convincing and compatible with the role of SYCE2 in the germ line (DSBs are formed in the Syce2 mutant and are

not efficiently processed). The data on the HP1a/SYCE2 interaction is less convincing, largely because it is based exclusively on the use of recombinant SYCE2.

Major points:

Figure 4: This figure leaves unanswered one important question: what are the HP1 foci we are looking at in panel C? Are they heterochromatin/H3K9me3 foci or sites of DNA repair? Panel S4E shows that some of the H3K9me3 foci become sites of HP1 accumulation upon depletion of SYCE2, while Figure 7D shows that foci of ATMp are mostly outside of H3K9me3 foci. This would support the model proposed by the authors (SYCE2 prevents HP1 from binding to H3K9me3 foci). Yet, the data from Figures 1 to 3 and from the world of meiosis, strongly suggest that in the absence of SYCE2, DSBs are not properly repaired and therefore accumulate. This is likely also to generate foci of HP1. To sort this out, the authors need to carry out a double labeling with HP1a and gH2Ax. Another concern in panel C is that there seems to be less HP1a in the foci-free cells in the siCont, and clearly less than in the siSYCE2. This difference is not seen in the control western in S4F, and could suggest that the MCF7 cells are heterogeneous, containing cells with foci and others without. An image of a larger field may allow sorting this out.

Figure 5: Panel A: HP1a migrates very close to the immunoglobulin light chain. The authors must show a larger part of their gel to convince the reader that he is not just looking variations in antibody concentrations. Panels G-I: Deletion of the hydrophobic region (rather than point mutations preserving the hydrophobicity) may impact on the overall structure of the protein. This should be taken into account when interpreting the loss of the interaction.

Figure 6: This figure shows that in the presence of SYCE2, it is no longer possible to detect H3K9me3 in the immunoprecipitate obtained with an anti-HP1a antibody. It is puzzling that the western in 6A shows a complete disappearance of the H3K9me3/HP1a interaction, while Figure 4F shows that in the RPE-FLAG-SYCE2 cells, there are still approx. 20% of the HP1a-domains that co-localize with H3K9me3. How do the authors explain this? In addition, although the data is clear, the interpretation may be complex. Is the histone H3 detected in the western part of a nucleosome or could SYCE2 be sequestering just free histone H3. This could be sorted out by an extractability assay (does SYCE2 make HP1a and H3 segregate in different fractions?).

Figure 7: Panel C: The use of the mutant deleting the hydrophobic domain is again problematic, as this mutant may not fold properly. The authors must at least show that this mutant retains the same sub-cellular localization as the WT (mostly nuclear).

Minor points

- "Mitotic cells" as opposed to "meiotic cells" is confusing. "Mitotic cells" may be understood by many as designating cells in mitosis (as opposed to interphase). I suggest using "somatic cells" as opposed to "germ line".
- The data presented in Panels 1A and 1B is essentially already available in Protein Atlas (<https://www.proteinatlas.org/ENSG00000161860-SYCE2/cell>) and may not need to be shown in a main figure.
- Fig S4B-D: HEK293 cells are indeed not coming from a tumor, but they are transformed with Adenovirus E1A and E1B.
- Some typos ("binded" in the caption of the model of Fig.7).

Reviewer #3 (Comments to the Authors (Required)):

SYCE2 is a synaptonemal complex component, generally thought to function exclusively in meiosis. In their paper "Mitotic role of SYCE2: an insulator that dissociates HP1 α from H3K9me3 and potentiates DNA repair" Hosoya et al study the role of SYCE2 in mitotic cells - and specifically in double strand break repair. Their major findings:

- 1) SYCE2 is expressed in cancer or transformed cell lines at varying levels. Its expression is not detectable in primary normal tissues.
- 2) Using a combination of siRNA knock down of SYCE2 in cells that do express it, and ectopic expression of the protein from a plasmid in cells that do not, the authors nicely show that expression of SYCE2 results in an improved double strand break response and higher repair by both NHEJ and HRR.
- 3) The authors show that while SYCE2 expression does not alter H3K9me3, it prevents the formation of HP1 α domains, resulting in more ATM activation and lower sensitivity to DSB and cisplatin.
- 4) The authors cloned deletion mutants for both HP1 α and SYCE2 to show interaction between

the proteins both in cells (by Co-IP) and in-vitro. They could further pin-point the interaction to the chromoshadow domain of HP1alpha and the N terminus of SYCE2.

In general, I find the experiments to be thorough and to well-support the authors conclusions that SYCE2 can promote double strand break repair in cancer cells through its interaction with HP1alpha.

Major comment:

- 1) The organization of the introduction is a bit unclear. There's an introduction into the synaptonemal complex, then into double strand breaks and HP1 - but it's not clear why the authors decided to study the connection between the two.
- 2) SYCE2 is a protein specifically expressed in cancer cells as opposed to normal cells. Furthermore - it confers better response and repair to double strand break inducing agents. This makes SYCE2 a prime-candidate for cancer selective therapy - a fact that should be emphasized to show the relevance of the work.
- 3) The analysis of foci throughout the paper is of "foci positive cells". This is a valid approach, but there could be added value to also counting the number of foci per cell which may also differ and be biologically relevant. In figure 7B it is stated that cells with "more than 3" foci were counted. Is this a different analysis than other figures? Why was this approach taken here as opposed to simply counting "foci positive" cells?

Minor comments:

- 1) Cisplatin forms both inter- and intra-strand crosslinks, and in fact, intrastrand crosslinks are much more abundant.
- 2) There was a previous paper in which SYCE2 was knocked out in mice and showed that this knockout also resulted in less efficient double strand break in meiosis. This paper should probably be quoted in the text: Bolcun-Filas E. et al, JCB 2007, DOI: 10.1083/jcb.200610027
- 3) I would avoid using too many acronyms, especially for names used only once or twice in the text. For instance, the use of TF for transversal filament - when many people connect TF with transcription factor.
- 4) Figure 7E - the model - is unclear. The two panels on the right and left are identical and seem redundant.
- 5) p.19, end of second line: preciously should be previously.

1st Revision – authors' response

12 May 2018

Responses to Reviewer #1

In this manuscript, the authors have investigated the function of SYCE2 in somatic cells. This protein was initially described as a subunit of the synaptonemal complex formed between homologous chromosomes during meiotic prophase. Yet, mRNA from this gene is also detected in several tissue culture cells and in some tumors. The authors convincingly show that knock-down of this mRNA in MCF7 cells results in decreased resistance to DNA-damaging agents, increased number of foci of gammaH2Ax, decreased phosphorylation of ATM at S1981, decreased RAD51 foci formation, and decreased 53BP1 foci formation (but only in irradiated cells). The opposite was seen upon forced expression of FLAG-SYCE2 in RPE cells. The authors then provide some data suggesting that SYCE2 interferes with HP1a foci formation by interacting with its dimerization (chromoshadow) domain. This interaction is not mediated by a classical PXXVL motif in SYCE2 but is abrogated by deletion of a hydrophobic domain of the protein. It is proposed that the HP1a/SYCE2 interaction prevents HP1a to interact with H3K9me3.

Evidence for the presence of SYCE2 transcript and protein outside the germ line is to be found on the Protein Atlas. Yet, this work seems to be the first to investigate the function of SYCE2 in somatic cells. The demonstration of a role for this protein in DDR is very convincing and compatible with the role of SYCE2 in the germ line (DSBs are formed in the Syce2 mutant and are not efficiently processed). The data on the HP1a/SYCE2 interaction is less convincing, largely because it is based exclusively on the use of recombinant SYCE2.

Major points:

[Comment]

Figure 4: This figure leaves unanswered one important question: what are the HP1 foci we are looking at in panel C? Are they heterochromatin/H3K9me3 foci or sites of DNA repair? Panel S4E shows that some of the H3K9me3 foci become sites of HP1 accumulation upon depletion of SYCE2, while Figure 7D shows that foci of ATMp are mostly outside of H3K9me3 foci. This would support the model proposed by the authors (SYCE2 prevents HP1 from binding to H3K9me3 foci). Yet, the data from Figures 1 to 3 and from the world of meiosis, strongly suggest that in the absence of SYCE2, DSBs are not properly repaired and therefore accumulate. This is likely also to generate foci of HP1. To sort this out, the authors need to carry out a double labeling with HP1a and γ H2Ax. Another concern in panel C is that there seems to be less HP1a in the foci-free cells in the siCont, and clearly less than in the siSYCE2. This difference is not seen in the control western in S4F, and could suggest that the MCF7 cells are heterogeneous, containing cells with foci and others without. An image of a larger field may allow sorting this out.

[Response]

We thank the referee for suggesting further experiments to answer an important question “Are the HP1 domains heterochromatin/H3K9me3 foci or sites of DNA repair?”. As suggested by the referee, we performed a double staining immunofluorescence study using antibodies for HP1 α and γ H2AX to test whether the HP1 α domains are colocalized with the γ H2AX foci induced upon 1 Gy irradiation. The percentages of HP1 α domains colocalized with γ H2AX were very small; $5.3 \pm 0.58\%$ in MCF7 cells treated with non-targeting control siRNA and $4.3 \pm 0.58\%$ in MCF7 cells treated with siRNA targeting SYCE2, showing no significant differences between these two types of cells ($p > 0.05$). The results indicate that the large HP1 α domains are located at heterochromatin (=H3K9me3 domains) rather than the sites of DNA repair. In the revised manuscript, we explained this in page 17, lines 4-13. We also added the sentences “Then, what are the large HP1 α domains observed in these figures? Are they heterochromatin (=H3K9me3 domains) or the sites of DNA repair? To sort this out, we ...” in page 16, lines 20-23, and also added a new supplementary Figure S4I, showing localization of HP1 α and γ H2AX in MCF7 cells knocked down with non-targeting control siRNA or siRNA for SYCE2.

As for the other concern about the HP1 α localization in MCF7 cells raised by the referee, the MCF7 cells are actually heterogeneous, containing cells with foci and others without. According to the referee’s request, we added images of larger fields in the new Figure S4B, and added a sentence “Immunofluorescence images of large fields stained with anti-HP1 α antibody suggest that MCF7 cells are heterogeneous, containing both cells with large HP1 α domains and others without (Fig S4B).” in page 16, lines 6-8, in the revised manuscript.

[Comment]

Figure 5: Panel A: HP1a migrates very close to the immunoglobulin light chain. The authors must show a larger part of their gel to convince the reader that he is not just looking variations in antibody concentrations. Panels G-I: Deletion of the hydrophobic region (rather than point mutations preserving the hydrophobicity) may impact on the overall structure of the protein. This should be taken into account when interpreting the loss of the interaction.

[Response]

We agree that HP1 α would possibly migrate very close to the immunoglobulin light chain. However, in this experiment, we performed immunoprecipitation using an anti-FLAG antibody produced in “mouse” or normal “mouse” IgG and subsequently performed Western blot analysis using the anti-HP1 α antibody produced in goat (not mouse) as a primary antibody and an HRP-linked anti-goat IgG antibody produced in rabbit as a secondary antibody. Using this condition, we could successfully avoid the problem of the detection of annoying bands of immunoglobulin light chain at 25kDa, making the bands for HP1 α easily detectable. In the revised manuscript, we added the information about the different species of the antibodies used in immunoprecipitation and Western blot analysis in the legend for the new Figure 5A (page 49, lines 4-6). According to the referee’s suggestion, we also replaced the old Figure 5A with a new one, which shows larger parts of the gel including the region surrounding the band of HP1 α .

We agree that it may be possible that the deletion of the hydrophobic region (rather than point mutations) may impact on the overall structure of the protein and that it should be taken into account when interpreting the loss of the interaction. Actually, we confirmed that the deletion mutant of FLAG-SYCE2 lacking the hydrophobic region shows the same subcellular localization as the wild type protein (mostly nuclear), which we added as a new supplementary Figure S7C in the revised manuscript. In addition, although we were told by Dr. Andrea Leibfried, the executive editor of Life Science Alliance, that additional mutations in SYCE2 (referee #1, comment regarding Figure 5) are not needed for acceptance in Life Science Alliance, we also constructed a “FLAG-SYCE2-49-56

mutant” harboring point mutations resulting in the substitution of the hydrophobic LTVLEGKS sequence into a non-hydrophobic ETDEEENS sequence, and found that this mutant also showed a remarkable reduction in the interaction with the chromoshadow domain of HP1 α . We believe that this additional data strongly supports and confirms the importance of the hydrophobic amino acids in the LTVLEGKS sequence of SYCE2 for the interaction with the chromoshadow domain of HP1 α . In the revised manuscript, we added this data of “FLAG-SYCE2-49-56 mutant” harboring point mutations in the new Figure 5J, and also added the explanation for this result in page 19, lines 8-12.

[Comment]

Figure 6: This figure shows that in the presence of SYCE2, it is no longer possible to detect H3K9me3 in the immunoprecipitate obtained with an anti-HP1 α antibody. It is puzzling that the western in 6A shows a complete disappearance of the H3K9me3/HP1 α interaction, while Figure 4F shows that in the RPE-FLAG-SYCE2 cells, there are still approx. 20% of the HP1 α -domains that co-localize with H3K9me3. How do the authors explain this? In addition, although the data is clear, the interpretation may be complex. Is the histone H3 detected in the western part of a nucleosome or could SYCE2 be sequestering just free histone H3? This could be sorted out by an extractability assay (does SYCE2 make HP1 α and H3 segregate in different fractions?).

[Response]

While we think that Figure 6A indicates a significant reduction in the HP1 α /H3K9me3 interaction in FLAG-SYCE2-expressing RPE cells, we do not think that SYCE2 expression may result in “complete” loss of the interaction. We think that immunofluorescence (Figure 4F) could more sensitively detect and reflect the remaining HP1 α /H3K9me3 interaction as “co-localization of the two signals” in FLAG-SYCE2-expressing RPE cells than the comparison of signal intensities in Western blot analysis where the sensitivity of the antibody will also affect the detection of the signals. We consider that inhibition of the HP1 α /H3K9me3 interaction occurs in a manner depending on the expression levels of SYCE2, as suggested by the result in Figure 6C.

As for the additional point, we thank the referee for suggesting an extractability assay to test whether SYCE2 makes HP1 α and histone H3 (or H3K9me3) segregate in different fractions. As suggested, we performed a salt-extractability assay in mock cells and RPE cells stably expressing FLAG-SYCE2, which is expected to differentially extract proteins according to their binding affinities to chromatin or nucleosome. As shown in the new supplementary Figure S4A, in this assay, we extracted the nuclear fraction in three steps by increasing the concentrations of NaCl in the extraction buffer (0.5M NaCl, 1.5M NaCl, and 2.5M NaCl). We show the results in the new supplementary Figure S4B in the revised manuscript. The H3K9me3 proteins were mostly extracted at a higher salt concentration of 1.5M NaCl in mock cells, confirming that most parts of H3K9me3 would be tightly incorporated into chromatin. On the other hand, histones H3 were mainly extracted at 1.5M NaCl (in the same fraction as H3K9me3), but a smaller pool of the histones H3 was also extracted at 0.5M NaCl in mock cells, which may reflect the existence of free histones, either newly synthesized or ejected from chromatin, that are not incorporated into chromatin. The extraction patterns of H3K9me3 and histone H3 were similar in mock cells and in RPE cells expressing FLAG-SYCE2, which may support the findings that SYCE2 expression does not affect the distributions of H3K9me3 (Figs 4B and S4A). Unexpectedly, HP1 α proteins were mainly extracted at a lower salt concentration of 0.5M NaCl and only a small pool of HP1 α proteins was extracted at 1.5M NaCl (in the same fraction as H3K9me3) even in mock cells. This suggests that the binding of HP1 α to H3K9me3 to form heterochromatin may be loose and that most parts of the HP1 α proteins are not tightly incorporated into chromatin even in mock cells showing very low expression of SYCE2. No differences in the extraction patterns of HP1 α were detected in mock cells and RPE cells expressing FLAG-SYCE2, indicating that dissociation of HP1 α from heterochromatin is hardly detectable by this salt-extractability assay. FLAG-SYCE2 in RPE cells was extracted at 0.5M NaCl, as expected from its diffuse localization in the nucleus. To conclude the results of the salt-extractability assay, we can at least say that SYCE2 does not affect the salt-extractability of either H3K9me3 at the nucleosome or total histone H3 including free histone H3. We added an explanation in page 20, lines 9- and page 21, lines 1-13.

[Comment]

Figure 7: Panel C: The use of the mutant deleting the hydrophobic domain is again problematic, as this mutant may not fold properly. The authors must at least show that this mutant retains the same sub-cellular localization as the WT (mostly nuclear).

[Response]

We agree with this comment. According to the suggestion, we examined the subcellular localization of the FLAG-SYCE2 mutant deleting the hydrophobic domain and found that this mutant is localized in the nucleus. Regarding this, we added a new supplementary Figure S7C and added a new sentence “We confirmed by immunofluorescence using anti-FLAG antibody that this mutant is located mostly in the nucleus, retaining the same subcellular localization as the wild-type SYCE2 protein (Figs S7C and 4A).” in the “Result” section (page 22, lines 19-21) of the revised manuscript.

Minor points:

[Comment]

- "Mitotic cells" as opposed to "meiotic cells" is confusing. "Mitotic cells" may be understood by many as designating cells in mitosis (as opposed to interphase). I suggest using "somatic cells" as opposed to "germ line".

[Response]

We thank the reviewer for the suggestion. According to the suggestion, we changed “mitotic cells” into "somatic cells" and “meiotic cell” into "germ line" in the revised manuscript (indicated in red).

[Comment]

- The data presented in Panels 1A and 1B is essentially already available in Protein Atlas (<https://www.proteinatlas.org/ENSG00000161860-SYCE2/cell>) and may not need to be shown in a main figure.

[Response]

We thank the reviewer for the suggestion. According to the suggestion, we moved the data presented in the old Figure 1A, B, and C to the supplementary Figure S1 in the revised manuscript.

[Comment]

- Fig S4B-D: HEK293 cells are indeed not be coming from a tumor, but they are transformed with Adenovirus E1A and E1B.

[Response]

We added the phrase “human embryonic kidney (HEK293) cells, which are not coming from tumors but are transformed with adenovirus type 5 E1A and E1B,” in page 8, lines 15-16.

[Comment]

- Some typos ("binded" in the caption of the model of Fig.7).

[Response]

We corrected the typographic errors (“binded”) to “bound” in the legend for Figure 6C (page 51, line 11) and in the legend for Figure 7E (the last line on page 51).

Responses to Reviewer #3

SYCE2 is a synaptonemal complex component, generally thought to function exclusively in meiosis. In their paper "Mitotic role of SYCE2: an insulator that dissociates HP1 α from H3K9me3 and potentiates DNA repair" Hosoya et al study the role of SYCE2 in mitotic cells - and specifically in double strand break repair. Their major findings:

- 1) SYCE2 is expressed in cancer or transformed cell lines at varying levels. It's expression is not detectable in primary normal tissues.
- 2) Using a combination of siRNA knock down of SYCE2 in cells that do express it, and ectopic expression of the protein from a plasmid in cells that do not, the authors nicely show that expression of SYCE2 results in an improved double strand break response and higher repair by both NHEJ and HRR.
- 3) The authors show that while SYCE2 expression does not alter H3K9me3, it prevents the formation of HP1 α domains, resulting in more ATM activation and lower sensitivity to DSB and cisplatin.
- 4) The authors cloned deletion mutants for both HP1 α and SYCE2 to show interaction between the proteins both in cells (by Co-IP) and in-vitro. They could further pin-point the interaction to the chromoshadow domain of HP1 α and the N terminus of SYCE2.

In general, I find the experiments to be thorough and to well-support the authors conclusions that SYCE2 can promote double strand break repair in cancer cells through its interaction with HP1 α .

Major comments:

[Comment]

1) The organization of the introduction is a bit unclear. There's an introduction into the synaptonemal complex, then into double strand breaks and HP1 - but it's not clear why the authors decided to study the connection between the two.

[Response]

1) We feel sorry that there was not a sufficient explanation indicating why we decided to study the connection between the synaptonemal complex protein and the double strand break repair in the original manuscript. We previously reported that another synaptonemal complex protein SYCP3 interferes with the BRCA2 tumor suppressor and inhibits the intrinsic homologous recombination pathway (Hosoya *et al*, EMBO rep 2012), which made us hypothesize that other synaptonemal complex proteins may also affect the DNA damage response and repair. In the revised manuscript, we added the following sentences at the end of the first paragraph in the “introduction” section (page 4, lines 15-20 in the first paragraph); “The roles of synaptonemal complex proteins in somatic cells are not well understood, except for the role of SYCP3 reported by our group (Hosoya *et al*, 2012). We reported that SYCP3 interferes with the BRCA2 tumor suppressor and inhibits the intrinsic homologous recombination pathway, indicating the role of a synaptonemal complex protein in regulating the DNA damage response and repair of DNA double-strand breaks (DSBs).” We believe that the connection between the first paragraph (an introduction into the synaptonemal complex) and the second paragraph (an introduction into the DNA damage response and repair) becomes much clearer.

[Comment]

2) SYCE2 is a protein specifically expressed in cancer cells as opposed to normal cells. Furthermore - it confers better response and repair to double strand break inducing agents. This makes SYCE2 a prime-candidate for cancer selective therapy - a fact that should be emphasized to show the relevance of the work.

[Response]

2) We thank the referee for giving us an important suggestion. According to the referee's suggestion, we emphasized the fact that SYCE2 is a prime-candidate for cancer selective therapy in the revised manuscript. We added the sentence “Taken together, these findings suggest that SYCE2 might be a prime candidate target for treatment of SYCE2-expressing tumors.” in the last paragraph of the “Introduction” section (page 7, lines 22-24). Secondly, we added the sentence “As SYCE2 is expressed at varying elevated levels in cancer cells as opposed to normal cells, SYCE2 might be a prime candidate target for cancer selective therapy.” in the first paragraph of the “Discussion” section (page 23, lines 11-13 in the first paragraph of the “Discussion” section).

[Comment]

3) The analysis of foci throughout the paper is of "foci positive cells". This is a valid approach, but there could be added value to also counting the number of foci per cell which may also differ and be biologically relevant. In figure 7B it is stated that cells with "more than 3" foci were counted. Is this a different analysis than other figures? Why was this approach taken here as opposed to simply counting "foci positive" cells?

[Response]

3) We actually adopted the same scoring methods of counting "foci-positive cells" throughout the paper, but the definitions for “foci-positive cells” were different among the molecules. For analysis of foci of γ H2AX, RAD51, and BRCA1, cells with "more than five (5)" large foci were scored as “foci-positive” cells. For analysis of foci of autophosphorylated ATM and 53BP1, cells with "three (3) or more large foci” were actually scored as “foci-positive” cells throughout the paper. We are sorry that there were mistakes in the descriptions of the legends for Figures 2D, 2E, 3D, 3E, S3C, 7B and 7C, analyzing foci of autophosphorylated ATM foci or 53BP1 in the original manuscript. In the revised manuscript, we corrected all the mistakes in these legends for figures, replacing all the mistaken descriptions of “more than three (3)” with "three (3) or more”. The method to analyze foci formation of autophosphorylated ATM in Figure 7B (and 7C) is not different from those in other figures (Figures 3D, 3E), where cells with “three (3) or more” large foci of autophosphorylated ATM are counted.

The reason why this approach (scoring cells with "three (3) or more” large foci as “foci-positive” as opposed to simply counting "foci-positive" cells) was taken to analyze foci formation of autophosphorylated ATM (and 53BP1) is as follows. The analysis of foci formation of

autophosphorylated ATM was performed in untreated and low-dose irradiated conditions (0.1-0.2 Gy). Large foci of autophosphorylated ATM are also occasionally visible in untreated cells, but the number of large foci in untreated mock cells is usually less than three (3). On the other hand, while large foci of autophosphorylated ATM can be induced upon X-ray irradiation, the numbers of large foci per cell induced by “low-dose” irradiation are small compared to those induced by high-dose irradiation of more than 1 Gy. Thus, the number of “3 or more” is valid to analyze the induction of autophosphorylated ATM foci in a low-dose irradiated condition as in the current case.

We agree that scoring “the number of foci per cell” may also be another valid and sensitive approach to analyze the foci formation. Thus, in the first step of analysis of foci formation, we also scored the numbers of foci per cell in a separate analysis to check whether the result obtained by counting the “percentage of foci-positive cells” in the present study correlates well with that obtained by scoring “the average number of foci per cell”. For example, in the analysis of foci formation of autophosphorylated ATM, we also scored the number of foci per cell in mock cells and RPE cells expressing FLAG-SYCE2. Expression of SYCE2 significantly increased the average number of autophosphorylated ATM foci per cell from 1.6 ± 0.34 in mock cells to 2.7 ± 0.36 in FLAG-SYCE2-expressing RPE cells in a steady state (in an untreated condition) ($p < 0.05$) and from 4.4 ± 0.43 in mock cells to 5.7 ± 0.19 in FLAG-SYCE2-expressing RPE cells upon 0.1 Gy X-ray irradiation ($p < 0.01$), which correlates well with the result of Figure 3E obtained by scoring the “percentages of foci-positive cells”.

Minor comments:

[Comment]

1) Cisplatin forms both inter- and intra-strand crosslinks, and in fact, intrastrand crosslinks are much more abundant.

[Response]

1) We revised the description in page 9, line 12 according to the referee’s comment; “We measured the ability of the cells to form colonies following exposure to ionizing radiation or cisplatin which produces both interstrand and intrastrand crosslinks.”

[Comment]

2) There was a previous paper in which SYCE2 was knocked out in mice and showed that this knockout also resulted in less efficient double strand break in meiosis. This paper should probably be quoted in the text: Bolcun-Filas E. et al, JCB 2007, DOI: 10.1083/jcb.200610027

[Answer]

2) According to the referee’s comment, we cited the paper in the text and revised the first sentence in the last paragraph of the “Introduction” section as follows; “In this study, we investigated the somatic roles of SYCE2 which have not been elucidated yet, while a previous paper reported that knockout of SYCE2 in mice resulted in less efficient double strand break repair in the germ line (Bolcun-Filas *et al*, 2007).” (page 7, lines 12-14).

[Comment]

3) I would avoid using too many acronyms, especially for names used only once or twice in the text. For instance, the use of TF for transversal filament - when many people connect TF with transcription factor.

[Answer]

3) According to the referee’s comment, we avoided to use the acronyms for axial/lateral elements (AEs), central element (CE), and transversal filaments (TFs) in the first paragraph of the “Introduction” section in the revised manuscript.

[Comment]

4) Figure 7E - the model - is unclear. The two panels on the right and left are identical and seem redundant.

[Answer]

4) According to the referee’s comment, we revised Figure 7E so that the contrast of DNA repair activities between the right panel (high SYCE2) and the left panel (low SYCE2) is clearly demonstrated.

[Comment]

5) p.19, end of second line: preciously should be previously.

[Answer]

5) We replaced the typographic error “preciously” with “previously”, which appears in page 25, line 2 in the revised manuscript.

2nd Editorial Decision

1 June 2018

Thank you for submitting your revised manuscript entitled "Somatic role of SYCE2: an insulator that dissociates HP1 α from H3K9me3 and potentiates DNA repair". The original reviewers assessed your work again.

As you will see, the reviewers now support publication. Reviewer #1 would have preferred an inclusion of the initial cytosolic/nuclear fractions in the blot for figure S6B. Should you have this data at hand, we would appreciate inclusion. But we are also happy to publish your paper in Life Science Alliance as is. Congratulations on this very nice work!

REFEREE REPORTS

Reviewer #1 (Comments to the Authors (Required)):

The extractability assay of the new figure S6B would have been more informative had it started with a cytosol/nuclear fractionation. But otherwise, the authors have properly and quite thoroughly addressed the issues that I raised.

Reviewer #3 (Comments to the Authors (Required)):

I am satisfied with how the authors addressed my comments and find the manuscript suitable for publication at Life Science Alliance.

2nd Revision – authors' response

3 June 2018

We appreciate your kind consideration to publish our paper in Life Science Alliance as is. While we agree that the extractability assay of the new figure S6B would have been more informative with an inclusion of the initial cytosolic/nuclear fractions in the blot as suggested by Reviewer #1, we unfortunately do not have this data at hand. Herewith we are sending our final files and also filled in the electron license to publish form.